# Identification of the SARS-unique domain of SARS-CoV-2 as an antiviral target

Bo Qin[1,2,6], Ziheng Li[1,2,6], Kaiming Tang [3,4,6], Tongyun Wang[4], Yubin Xie[4], Sylvain Aumonier[5], Meitian Wang [5], Shuofeng Yuan [3,4] ✉ & Sheng Cui [1,2] ✉

SARS-CoV-2 nsp3 is essential for viral replication and host responses. The SARS-unique domain (SUD) of nsp3 exerts its function through binding to viral and host proteins and RNAs. Herein, we show that SARS-CoV-2 SUD is highly flexible in solution. The intramolecular disulfide bond of SARS-CoV SUD is absent in SARS-CoV-2 SUD. Incorporating this bond in SARS-CoV-2 SUD allowed crystal structure determination to 1.35 Å resolution. However, introducing this bond in SARS-CoV-2 genome was lethal for the virus. Using biolayer interferometry, we screened compounds directly binding to SARS-CoV-2 SUD and identified theaflavin 3,3'-digallate (TF3) as a potent binder, $K_d$ 2.8 μM. TF3 disrupted the SUD-guanine quadruplex interactions and exhibited anti-SARS-CoV-2 activity in Vero E6-TMPRSS2 cells with an $EC_{50}$ of 5.9 μM and $CC_{50}$ of 98.5 μM. In this work, we provide evidence that SARS-CoV-2 SUD harbors druggable sites for antiviral development.

Three years into the Coronavirus Disease 2019 (COVID-19) pandemic, antiviral drugs authorized and the approved for treating COVID-19 remain limited. Whereas Remdesivir[1], Paxlovid[2], and Molnupiravir[3] target only two important virally encoded proteins, the main protease (M^pro) and the RNA-dependent RNA polymerase (RdRp), the genome of SARS-CoV-2 (~30 kb, the largest among RNA viruses[4]) encodes many proteins (~28 in total), providing a wealth of potential therapeutic targets that remain to be exploited.

Among all SARS-CoV-2 encoded proteins, the nonstructural protein 3 (nsp3) is the largest, and it participates in many essential steps in the virus life cycle. It's roles include polyprotein processing, replication compartment formation, replication-transcription complex formation, nascent viral RNAs trafficking between double membrane vesicles (DMV) and cytoplasm and innate immunity antagonism[5]. The 1,945 residues of SARS-CoV-2 nsp3 fold into at least 15 domains, including ubiquitin-like domains, acidic domains, macrodomains, papain-like protease (PLpro) domains and transmembrane domains, etc., (Fig. 1A), providing numerous potentially druggable sites within a single polypeptide chain[6]. Since the beginning of COVID-19 pandemics, antiviral agent development has largely focused on the main protease (Mpro) and the RNA-dependent RNA polymerase (RdRP), but most efforts targeting the nsp3 have been focused on its PLpro domain. Fragment-based drug development was employed to identify novel compounds targeting the CoV-conserved nsp3-macrodomain (also known as Mac1 or X domain)[7,8]. High-resolution structures of SARS-CoV-2 PLpro complexed with numerous inhibitors have been determined[9,10], revealing novel strategies for antiviral design.

In addition to the CoV-conserved domains of nsp3, SARS-CoV-2 nsp3 harbors a "SARS-Unique Domain" (SUD), which was first identified in the SARS-CoV genome[11]. SARS-CoV-2 SUD shares ~75% amino acid sequence identity with SARS-CoV SUD. As its name indicated, the SUD region had not been found in other less pathogenic CoVs at the time of its discovery; therefore, it was thought that SUD is responsible for pathogenesis. SUD contains three subdomains; there are two macrodomains, Mac2 and Mac3, that are followed by a frataxin-like Domain Preceding Ubl2 and PL2pro (DPUP). For simplicity, they are also known

[1]NHC Key Laboratory of Systems Biology of Pathogens, Institute of Pathogen Biology Chinese Academy of Medical Sciences & Peking Union Medical College, 100730 Beijing, China. [2]Key Laboratory of Pathogen Infection Prevention and Control (Peking Union Medical College), Ministry of Education, 100730 Beijing, China. [3]State Key Laboratory of Emerging Infectious Diseases, Li Ka Shing Faculty of Medicine, The University of Hong Kong, Pokfulam, Hong Kong SAR, China. [4]Department of Microbiology, Li Ka Shing, Faculty of Medicine, The University of Hong Kong, Pokfulam, Hong Kong SAR, China. [5]Swiss Light Source at the Paul Scherrer Institute, 5232 Villigen, Switzerland. [6]These authors contributed equally: Bo Qin, Ziheng Li, Kaiming Tang. ✉e-mail: yuansf@hku.hk; cui.sheng@ipb.pumc.edu.cn

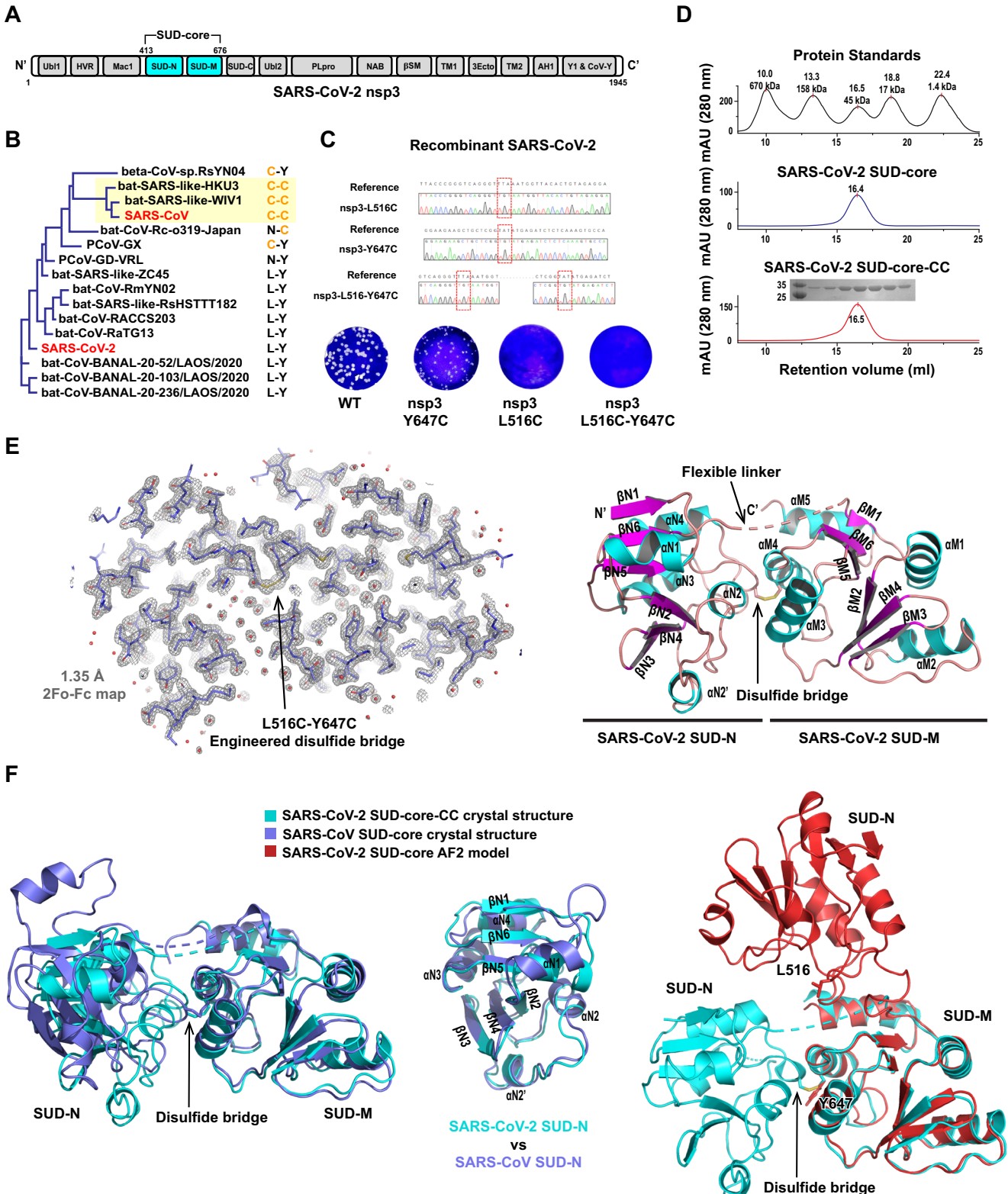

**A** SARS-CoV-2 nsp3

**B**

**C** Recombinant SARS-CoV-2

**D** Protein Standards / SARS-CoV-2 SUD-core / SARS-CoV-2 SUD-core-CC

**E** 1.35 Å 2Fo-Fc map / L516C-Y647C Engineered disulfide bridge / Flexible linker / SARS-CoV-2 SUD-N / SARS-CoV-2 SUD-M

**F**
- ■ SARS-CoV-2 SUD-core-CC crystal structure
- ■ SARS-CoV SUD-core crystal structure
- ■ SARS-CoV-2 SUD-core AF2 model

SARS-CoV-2 SUD-N vs SARS-CoV SUD-N

as SUD-N, SUD-M and SUD-C. After the emergence of SARS-CoV, domains homologous to SUD-M and SUD-C were identified not only in Sarbecoviruses including SARS-CoV-2, but also in other CoV lineages, e.g., in Middle East respiratory syndrome coronavirus and Mouse hepatitis virus[12–14]. By contrast, SUD-N remains unique to Sarbecoviruses.

Previous structural characterization of SARS-CoV SUD identified a stable and crystallizable fragment containing SUD-N and SUD-M[15], subsequently studied as an intact domain, denoted SUD-core. The crystal structure of SARS-CoV SUD-core revealed two macro-like domains SUD-N and SUD-M, connected by a highly flexible linker. Of note, an unusual disulfide bridge is formed between SUD-N and SUD-M, which may strengthen their connection. Intriguingly, the subcellular location of nsp3 is in the reductive cytoplasm, which does not support disulfide bridge formation (this requires an oxidative environment). Therefore, the function of this disulfide bridge demands further investigation.

**Fig. 1 | SARS-CoV-2 SUD-core lacks the disulfide bridge connecting SUD-N and SUD-M domains. A** Schematics of SARS-CoV-2 nsp3 (1945 aa) domain organization. A stable fragment of SARS-CoV-2 nsp3 (residue 413–676) was expressed and crystallized in this study, which contains SUD-N and SUD-M (colored cyan). **B** Phylogenetic tree of a selection of Sabecoviruses nsp3 sequences, shown with real branch length. A subbranch containing SARS-CoV nsp3 harboring a pair of cysteines allows disulfide bridge formation between SUD-N and SUD-M is highlighted with yellow background. SARS-CoV-2 nsp3 and SARS-CoV nsp3 are highlighted with red fonts. Disulfide-forming cysteines of SARS-CoV nsp3 (orange fonts) and their counterparts in different nsp3 are listed on the right. **C** Left, recombinant SARS-CoV-2 viruses harboring mutation nsp3-L516C and/or nsp3-Y647C were prepared from infectious clone pBAC-SARS-CoV-2. Top, DNA sequencing confirmed the indicated mutations in SARS-CoV-2 infectious clones; bottom, plaque assays of wild-type SARS-CoV-2 virus, the mutants harboring nsp3-Y647C, nsp3-L516C and nsp3-Y647C-L516C. **D** Size-exclusion chromatography of SARS-CoV-2 SUD-core and SUD-core-CC containing the disulfide bridge mutations L516C-Y647C. Top, profile of protein standards; middle, profile of SARS-CoV-2 SUD-core; bottom, profile of SARS-CoV-2 SUD-core-CC and an SDS-PAGE of the fractions eluted from size-exclusion column is aligned with the profile. **E** Crystal structure of SARS-CoV-2 SUD-core-CC was determined to 1.35 Å resolution. Left, stick model of SUD-core-CC is superimposed with the final 2Fo-Fc electron density map (gray mesh, 1.5 σ); right, ribbon model SUD-core-CC is colored by secondary structural elements (α-helix cyan, β-strand magenta and loop brown). Each secondary structural element is labeled. The engineered disulfide bridge L516C-Y647C formed between SUD-N and SUD-M is indicated by arrow. **F** Left, structural alignment of SARS-CoV-2 SUD-core-CC (cyan) and SARS-CoV SUD-core (PDB: 2W2G, blue), RMSD = 3.0 Å. Relative position varies between SUD-N and SUD-M in two structures, whereas the folding of the individual SUD-N and SUD-M remains similar (RMSD = 0.8 Å). Middle, Structural alignment of SARS-CoV-2 SUD-N with SARS-CoV SUD-N, RMSD = 1.7 Å. Right, structural alignment of SARS-CoV-2 SUD-core-CC (cyan) and the AlphaFold2 model of SARS-CoV2 SUD-core (red), RMSD = 5.2 Å; relative position of SUD-N and SUD-M is completely different in this model. The disulfide folding residues of SARS-CoV SUD are replaced by L516 and Y647 (shown with stick model) in SARS-CoV-2 SUD.

SUD-core does not possess enzymatic activity; rather, it functions as a binding module for viral and host molecules[16,17]. Unlike Mac1 or other known macrodomains, SUD-core does not bind ADP-ribose; instead, it binds various nucleic acids and proteins. SARS-CoV SUD-core recognizes a special type of nucleic acids that folds into guanine-quadruplex (G4) structures, which is essential for virus replication[13,18]. Similarly, SARS-CoV-2 SUD retains the ability to bind RNA/DNA G4[17]. Structural characterization of SARS-CoV-2 SUD-core remained lacking and bridging this knowledge gap is important for understanding of this pandemic pathogen. Computational modeling has suggested significant differences between the structures of SARS-CoV-2 SUD and SARS-CoV SUD, either alone or in complex with G4. SUD interacts with host proteins; both SARS-CoV-2 SUD and SARS-CoV SUD interact with human poly (A)-binding protein-interacting protein 1 (Paip1)[16], through which the viruses commandeer the cellular protein translation apparatus for their own benefits. The structure of SARS-CoV SUD-N complexed with Paip1 middle domain (Paip1M) has revealed that the N-terminal ~20 amino acids of the SUD-N domain interact directly with Paip1M. A larger nsp3 fragment containing SUD and PLpro domains binds the E3 ubiquitin ligase RCHY1 and their interaction is important for SARS-CoV virulence[14].

In the present study, we characterized SARS-CoV-2 SUD structurally and biochemically. By incorporating a SARS-CoV SUD-like disulfide bond, we stabilized SARS-CoV-2 SUD-core thus determined its crystal structure to 1.35 Å resolution. We introduced mutations into the SARS-CoV-2 SUD genomic sequence that resulted in a disulfide between the domains SUD-N and SUD-M and found such changes depleted virus viability. We compared SARS-CoV-2 and SARS-CoV SUDs with respect to thermostability, oligomerization state and binding status to the Paip1M in solution. As a proof-of-concept, we screened small molecule compound libraries and identified a group of antiviral hit compounds functioning by disrupting the SUD-G4 interaction. Our study hinted an avenue for developing next-generation antiviral agents targeting SARS-CoV-2 SUD.

## Results

### The unusual SARS-CoV SUD disulfide bridge is absent in SARS-CoV-2

SARS-CoV nsp3 and SARS-CoV-2 nsp3 share ~75% amino acid sequence identity. The unusual disulfide bond bridging SARS-CoV SUD-N and SUD-M (C492-C623) is absent in SARS-CoV-2 SUD, because the disulfide-forming cysteines are substituted with Leu and Tyr in SARS-CoV-2 (L516-Y647). Phylogenetic analyses among Sarbecoviruses found that this disulfide bond is only present in a small branch of viruses closely related to SARS-CoV, and the disulfide-forming cysteines have gradually mutated in other Sabecoviruses to hydrophobic amino acids in most cases (Fig. 1B and Supplementary Fig. 1).

The observation implies a possible evolutionary trend toward loss of this disulfide bond, which persuade us to investigate its role further. To this end, we mutated one of SARS-CoV-2 nsp3 residues L516 and Y647, or both, back to cysteine in a reverse genetic system (pBAC-SARS-CoV-2) for recombinant SARS-CoV-2, respectively (Fig. 1C). We found that L516C alone and double mutation L516C-Y647C are lethal to SARS-CoV-2 production, but Y647C alone is tolerable. The results indicate that the SARS-CoV SUD-like interdomain disulfide bond is incompatible with SARS-CoV-2 viability, presenting a key difference between SARS-CoV-2 and SARS-CoV.

### Structural characterization of the SARS-CoV-2 SUD-core

To reveal the structural basis underlying the function of SARS-CoV-2 SUD, we characterized its structure using crystallographic approaches. We expressed a nsp3 fragment (residue 413–676) comprising SUD-N and SUD-M domains, denoted SARS-CoV-2 SUD-core. Although this fragment was highly soluble (Fig. 1D), it failed to crystalize despite considerable efforts. We reasoned that the lack of disulfide bond between SUD-N and SUD-M might confer high flexibility and prevented the crystallization. Therefore, we engineered a disulfide bond in SARS-CoV-2 SUD-core based on SARS-CoV SUD-core; the resulting mutant containing double mutations L516C-Y647C was denoted SARS-CoV-2 SUD-core-CC. Clearly, the L516C-Y647C mutations did not affect the stability or oligomerization state of the protein (Fig. 1D).

To crystalize SARS-CoV-2 SUD-core-CC, we avoided using reducing reagents in latter stages of purification to promote disulfide bond formation. The protein was finally exchanged into an oxidative buffer and incubated overnight before crystallization screening. The crystals of SARS-CoV-2 SUD-core-CC belonged to the space group of P321 and the best crystals diffracted the X rays to 1.35 Å resolution. Initial SUD crystals diffracted to ~ 2.3 Å, and the crystals were gradually improved by optimizing crystallization conditions. After the screening of SUD-binding compounds (detailed in bellow sections), the SUD crystals co-crystallized with the hit compound 4 diffracted to the highest resolution 1.35 Å, thus was used for in depth analysis. The crystal structure of SARS-CoV-2 SUD-core-CC was solved by molecular replacement using SARS-CoV SUD-core structure as a searching model (PDB: 2W2G[15]). Because the N-terminal region of SUD-N (~20 amino acids) adopted a completely different fold from that of the searching model, this region was rebuilt manually. Statistics for data collection and refinement are summarized in Table 1.

SUD-core-CC comprises tandem SUD-N and SUD-M macrodomains connected by a flexible loop region. Both SUD-N and SUD-M adopt the typical 'macro fold', a mixed six-stranded β-plane is sandwiched by short α-helices and connecting loops on both sides (Fig. 1E, Supplementary Fig. 2). A nine-residue loop region between SUD-N and SUD-M (residues 541-549) is completely missing from the electron

**Table 1 | Data collection and refinement statistics**

| | SARS-CoV-2 SUD-core-CC Soaked with Comp.4 (PDB ID: 8GQC) | SARS-CoV-2 SUD-core-CC Protein alone (PDB ID: 8HBL) |
|---|---|---|
| **Data collection** | | |
| Space group | P321 | P321 |
| Cell dimensions | | |
| $a, b, c$ (Å) | 87.01, 87.01, 76.77 | 85.67, 85.67, 76.66 |
| $\alpha, \beta, \gamma$ (°) | 90, 90, 120 | 90, 90, 120 |
| Resolution (Å) | 75.35–1.35 (1.49–1.35) | 42.84–1.58 (1.68–1.58) |
| $R_{sym}$ or $R_{merge}$ | 0.04 (1.733) | 0.07 (1.75) |
| $I/\sigma I$ | 30.5 (1.70) | 21.99 (1.76) |
| Completeness (%) | 96.00 (72.10) | 99.60 (98.50) |
| Redundancy | 20.00 (19.35) | 19.86 (20.36) |
| **Refinement** | | |
| Resolution (Å) | 37.85–1.35 | 42.84–1.58 |
| No. reflections | 55,555 | 44,266 |
| $R_{work}/R_{free}$ | 0.14/0.16 | 0.14/0.18 |
| No. atoms | | |
| Protein | 3,889 | 3901 |
| Ligand/ion | | 30 |
| Water | 314 | 200 |
| B-factors | | |
| Protein | 32.98 | 40.81 |
| Ligand/ion | | 60.89 |
| Water | 43.04 | 45.84 |
| R.m.s. deviations | | |
| Bond lengths (Å) | 0.009 | 0.018 |
| Bond angles (°) | 1.062 | 1.627 |

[a]Values in parentheses are for highest-resolution shell.

density map despite its atomic resolution, suggesting that this interdomain linkage is highly flexible. We clearly observed the engineered disulfide bond between L561C and Y647C (Fig. 1E left), tethering SUD-N and SUD-M together. Conversely, the SARS-CoV-2 SUD-core lacking this bond may be free to undergo greater interdomain movement, which is probably essential for its function and the viability of SARS-CoV-2. We superimposed SARS-CoV-2 SUD-core-CC and SARS-CoV SUD-core, and 257 Cα atoms aligned with a Dali Z-score of 24.1, and a root mean square deviation (RMSD) of 2.6 Å. This high structural deviation stemmed from distinct interdomain orientations between SUD-N and SUD-M in the two SUD structures (Fig. 1F left), rather than differences in the individual domains themselves. The structure of the isolated SUD-N and SUD-M of SARS-CoV-2 remain similar to their counterparts in SARS-CoV SUD. The RMSD between the SUD-N of SARS-CoV-2 and SARS-CoV is 1.7 Å; the RMSD between the SUD-M of SARS-CoV-2 and SARS-CoV is 0.8 Å (Table S1). Additionally, structural differences were also seen near the N-terminus of SUD-N, at αN1-βN1 (Fig. 1F left).

Given that our crystal structure may not necessarily reflect the natural conformation of SARS-CoV-2 SUD-core due to the engineered disulfide bond in it, we used program AlphaFold2 (AF2) to predict its structure (Fig. 1F right). The highest ranked AF2 model had an overall predicted local distance difference test (pLDDT) value of 86.5, indicating a medium confidence of prediction. The pLDDT per residue plot (Supplementary Fig. 3) showed that while two macrodomains had the highest pLDDT value (90-100), the linker between them had the lowest values (<50); therefore, the prediction of the linker conformation was unreliable. Superimposition of the AF2 model and the crystal structure demonstrates that the relative position between SUD-N and SUD-M

varied greatly between the two models (Fig. 1F right, Supplementary Table 1). Surprisingly, residues L516 and Y674 are far apart in the AF2 model, suggesting that they do not interact with each other. Although there is insufficient evidence to judge which conformation is physiologically relevant, it is safe to assume that the linkage between SARS-CoV-2 SUD-N and SUD-M is highly flexible.

Next, we employed circular dichroism (CD) spectroscopy to investigate solution structural properties of SARS-CoV-2 SUD-core, SARS-CoV-2 SUD-core-CC and SARS-CoV SUD-core (Supplementary Fig. 4). Under oxidative conditions, the CD spectrum of SARS-CoV-2 SUD-core-CC is very different from that of SARS-CoV-2 SUD-core, although they are only different by two disulfate bridge forming residues. By contrast, the CD spectrum of SARS-CoV SUD-core containing a natural disulfate bridge is more similar to that of SARS-CoV-2 SUD-core-CC, despite there are ~ 25% difference between their sequences. Additionally, we measured melting temperatures (Tm) for these proteins. As expected, SARS-CoV-2 SUD-core exhibited lower Tm than that of the other two proteins, suggesting the disulfate bridge between SUD-N and SUD-M contributes to the overall SUD stability in solution.

### SARS-CoV-2 SUD-core and SARS-CoV SUD-core bind host Paip1M and G4 RNA with different affinities

We demonstrated that while SARS-CoV-2 SUD-N and SUD-M domains are flexibly linked, SARS-CoV SUD-N and SUD-M are fixed through a disulfide bond. To understand whether this distinctive feature affects SUD function, we compared the overall stability and the ability of the two Sarbecovirus SUD-cores to bind the human Paip1 middle domain or a host G4-RNA TRF2, a G4 motif present in the 5′-UTRs of TRF2 mRNA.

We measured the thermostability of SARS-CoV-2 SUD-core and SARS-CoV SUD-core in solution using thermal shift experiments (Fig. 2A) and calculated their melting temperature ($T_m$) to be 40 °C and 44 °C, respectively. The remarkably higher thermostability of SARS-CoV SUD-core is consistent with the presence of an internal disulfide bond that might restrict interdomain movement. Both SARS-CoV SUD-core and SARS-CoV-2 SUD-core bind Paip1M, a key component of cellular protein translation initiation complexes. The interfacial area between SUD-core and Paip1M was mapped between the N-terminal ~20 amino acids of SUD-N and the Paip1 middle HEAT repeat domain (Paip1M)[16]. We therefore purified SARS-CoV-2 SUD-core, SARS-CoV SUD-core and Paip1M (Supplementary Fig. 5) and confirmed that both SUD-cores formed stable complexes with Paip1M (Fig. 2B).

We then investigated the SUD-Paip1M interaction using bio-layer interferometry (BLI). Paip1M was immobilized on an AR2G biosensor to measure binding to SUD-core in solution (Fig. 2C left, Supplementary Fig. 6A–C). Paip1M exhibited a ~ 4 folds higher affinity for SARS-CoV SUD-core than for SARS-CoV-2 SUD-core with equilibrium dissociation constants $K_d$ value of 4.4 µM and 18.0 µM, respectively. Nevertheless, the N-terminal sequence of SUD-N in SARS-CoV-2 and SARS-CoV are nearly identical (Supplementary Fig. 2), which cannot explain their different affinities for Paip1M. We measured the binding of the mutant SARS-CoV-2 SUD-core-CC with Paip1M. The mutant harboring an engineered disulfide bridge between SUD-N and SUD-M exhibited higher affinity with Paip1M, suggesting the disulfide bridge plays an important role in Paip1M binding, although we cannot rule out the contribution of other SUD residues. To further understand the role of the disulfide bridge in other SUD functions, we also investigated the SUD-G4 RNA interaction using BLI (Fig. 2C right, Supplementary Fig. 6D–F). Whereas SARS-CoV SUD-core binds G4-TRF2 with a $K_d$ of 17 nM, SARS-CoV-2 SUD-core exhibited 10-fold weaker binding affinity to G4-TRF2 ($K_d$ =170 nM). To our surprises, binding affinity with G4-TRF2 was recovered when the disulfide bridge was introduced. SARS-CoV-2 SUD-core-CC binds G4-TRF2 with some 34-folds higher affinity ($K_d$ = 5 nM) than SARS-CoV-2 SUD-core, suggesting the disulfide bridge is also important to G4-RNA binding.

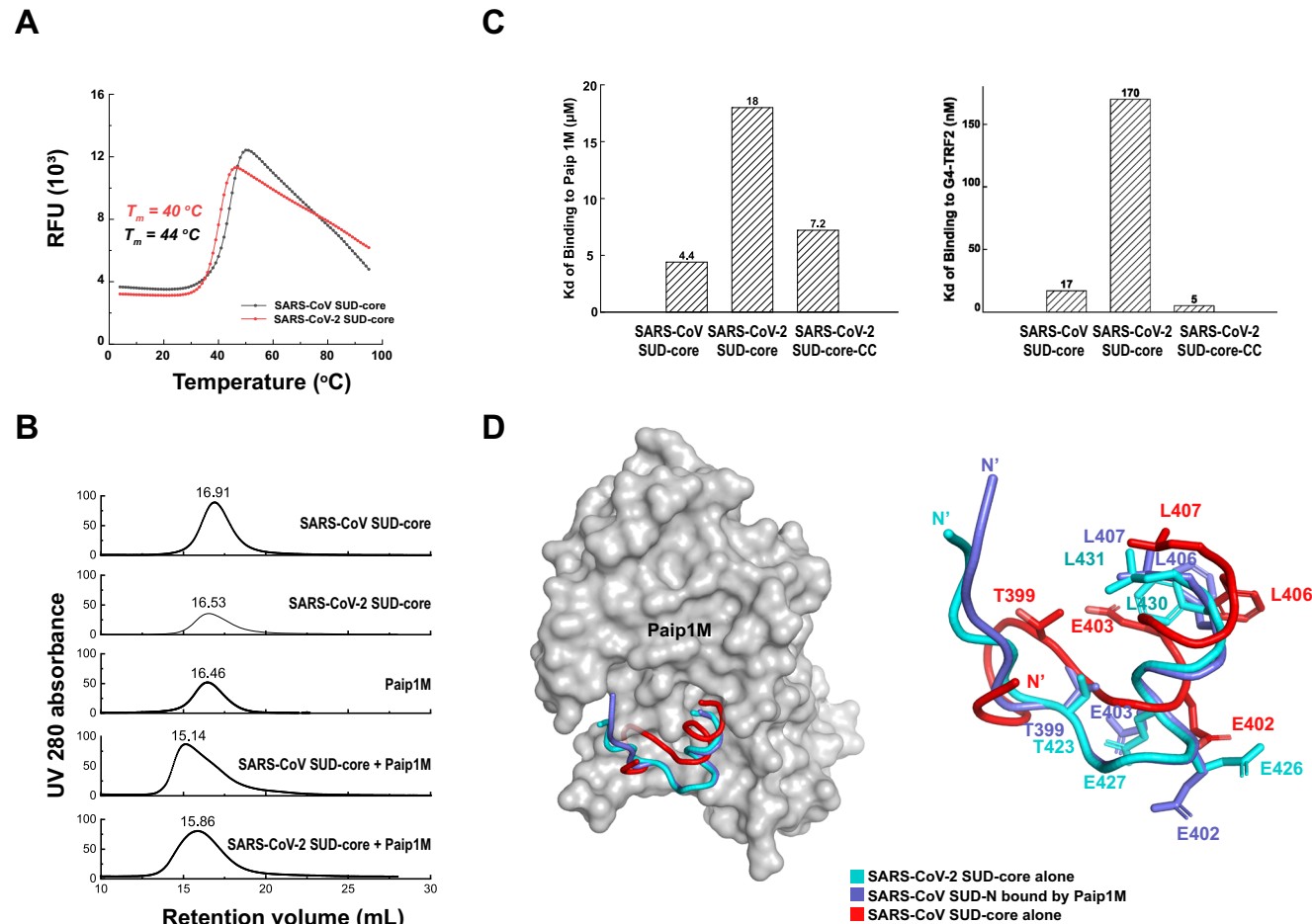

**Fig. 2 | SARS-CoV-2 SUD-core and SARS-CoV SUD-core interact with Paip1M with different affinity. A** Thermostability measurement of purified SUD-core proteins by thermal shift experiments; the melting curves of SARS-CoV SUD-core (black) and SARS-CoV-2 SUD-core (red) are shown. Their melting temperatures $T_m$ are indicated on the left. SARS-CoV SUD-core is more stable than SARS-CoV-2 SUD-core in solution. **B** SUD-core forms stable complex with Paip1M. Recombinant SARS-CoV-2 SUD-core and SARS-CoV SUD-core were incubated with Paip1M with 1:1.2 molar ration before loading to Superdex 200 10/300 GL. Both SARS-CoV-2 SUD-core: Paip1M and SARS-CoV SUD-core: Paip1M eluted as stable complexes from the size-exclusion column. **C** SUD-Paip1M binding and SUD-G4 RNA binding were investigated using BLI titrations. Purified Paip1M was immobilized on AR2G biosensor for binding with SARS-CoV SUD-core, SARS-CoV-2 SUD-core and SARS-CoV-2 SUD-core-CC; the $K_d$ values are represented as column graphs (right). G4-RNA TRF2 was immobilized on SA biosensor for binding with SARS-CoV SUD-core,

SARS-CoV-2 SUD-core or SARS-CoV-2 SUD-core-CC; the $K_d$ values are represented as column graphs (right). Details of the above BLI titrations including sensorgrams, binding kinetic parameters $K_d$, $k_{on}$, and $k_{off}$ are shown in Supplementary Fig. 6. **D** Left, the structure of SARS-CoV SUD-N complexed by Paip1M (PDB: 6YXJ[16]) reveals that the N-terminal ~20 amino acids of SUD-N (blue) is directly involved in the binding with Paip1M (gray). The structure of SARS-CoV-2 SUD-core (cyan, report in this study) and the structure of unbound SARS-CoV SUD-core (red, PDB: 2W2G[15]) are superimposed to the Paip1M-bound SARS-CoV SUD-N. Whereas Paip1M is shown with molecular surface, only the N-terminal loops from different SUD-N structures are shown with ribbon models. Right, magnified view of the super-imposed SUD N-terminal loops shown; SUD residues directly involved in the interaction with Paip1M, and their structural counterparts in SARS-CoV-2 SUD are shown with stick models and labeled.

To explore the mechanism for SUD-Paip1M binding, we super-imposed the SARS-CoV-2 SUD-core structure and the unbound SARS-CoV SUD-core structure onto the Paip1M-bound SARS-CoV SUD-core structure (Fig. 2D left). The N-terminal ~20 residues of SARS-CoV SUD-N adopt remarkably different conformations in the presence and absence of Paip1M, implying an 'induced-fit' mechanism for binding to Paip1M (Fig. 2D right). By contrast, The N-terminal loop of unbound SARS-CoV-2 SUD-N adopts a fold highly similar to that of the Paip1M-bound SARS-CoV SUD-N. Therefore, the N-terminal loop of SARS-CoV-2 SUD-N adopts a 'ready-to-bind' conformation, implying a 'lock-and-key' mechanism for Paip1M binding. Collectively, the difference in SUD-Paip1-binding affinity between SARS-CoV and SARS-CoV-2 is probably attributed to their distinct binding mechanisms, which stems from their different internal structural features, rather than the N-terminal residues of SUD-N at the binding interface.

## Analysis of key residues involved in SUD-G4 interaction

Many protein-nucleic acid interactions are electrostatic interactions. SUD harbors positively charged surfaces, which might serve as putative G4-binding sites[17]. Therefore, we screened surface charged residues of SARS-CoV-2 SUD for their contributions to G4 binding. We selected both positively and negatively charged residues for mutagenesis, and prepared SUD mutants bearing single and combined alanine substitutions. Stability of the mutants were assessed using SDS-PAGE analysis and thermal shift assays. As shown in Supplementary Fig. 7, SUD mutants were expressed and purified, and their $T_m$ values were all above 25 °C, the temperature used for BLI titration. We then measured the binding affinity of the mutants to G4-TRF2 using BLI (Fig. 3A, B). Combined mutations were typically more effective at disrupting G4-binding than single mutations. Among the single mutations, D448A and E595A had a negligible effect on binding affinity but all others impaired G4-binding to

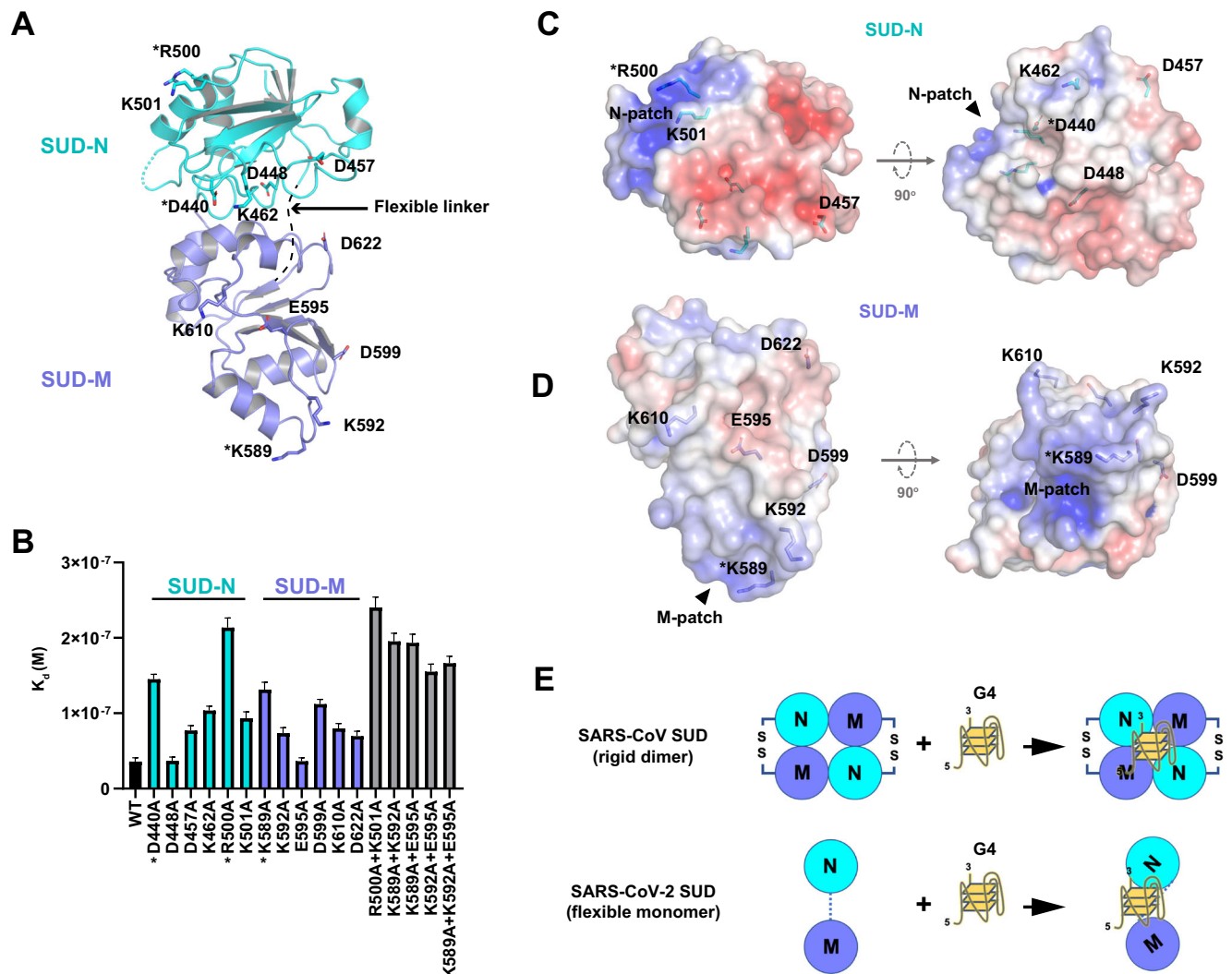

**Fig. 3 | Mapping SARS-CoV-2 SUD residues important for the SUD-G4 interaction. A** SARS-CoV-2 SUD-core structure is shown with ribbon model and colored by domains, SUD-N colored cyan, SUD-M colored blue. A selection of charged residues on the surface of SUD-core are shown with stick model and indicated. The region between SUD-N and SUD-M is highly flexible and missing from electron density, therefore indicated by a dashed line. **B** SARS-CoV-2 SUD-core mutants containing alanine substitution of the charge residues shown in panel **A** were prepared. The binding affinity of SUD mutants with G4-TRF2 was measured by BLI; dissociation constant ($K_d$) of the mutants is compared with WT SUD. All data were shown as mean ± SEM, $n$ = 2 biologically independent experiments. **C, D** Surface electrostatic potential plot (blue, positive charge; red, negative charge) of molecular surface of the individual SARS-CoV-2 SUD-N (panel **C**) and SUD-M (panel **D**). Left, side view (the same orientation as in panel **A**). Right, bottom view. Residues picked for mutagenesis study are shown with stick models and indicated. The predominantly positively charged patches on SUD-N (N-patch) and SUD-M (M-patch) are indicate. Star indicates residue important to the SUD-G4 binding. **E** Schematics showing different mechanism in SUD-G4 interaction; SUD-N is shown with cyan circles, SUD-M is shown with blue circles, dashed line indicates flexible linker between domains. Previous studies showed that SARS-CoV SUD forms rigid dimer stabilized by dimerization interface and disulfide bonds; the predicted G4-binding site lies between SARS-CoV SUD monomers. SARS-CoV-2 forms monomers in solution and it lacks interdomain disulfide bonds. SARS-CoV-2 SUD-N and SUD-M are connected by a flexible linker, and they both participate in G4-binding.

various degrees. Because mutations of both SUD-N and SUD-M impaired G4-binding affinity, the binding site of G4 is probably located between the two domains.

We performed surface electrostatic potential analysis of the SARS-CoV-2 SUD-core-CC structure. Because wild type (WT) SUD does not contain an interdomain disulfide bond, SUD-N and SUD-M may not necessarily interact with each other like they do in crystals. Therefore, we analyzed the surface electrostatic potential of the isolated SUD-N and SUD-M (Fig. 3C, D). We showed that each macrodomain of SUD harbors a positively charged patch (denoted N-patch and M-patch), suggesting that they are implicated in G4 binding. Consistent with this prediction, R500A and K598A that caused a severe loss in G4-binding affinity (Fig. 3B) are located at N-patch and M-patch, respectively. Although the structure of SARS-CoV-2 SUD-core-CC indicates that N-patch and M-patch are far apart, this structure may not reflect the

physiologically relevant conformation of WT SUD lacking the engineered disulfide bond. Our structural and biochemical results indicate that SARS-CoV-2 SUD-N and SUD-M are connected by a flexible linker, which may allow enough freedom for SUD-N and SUD-M to adopt an optimal conformation to reach G4 simultaneously (Fig. 3E). By contrast, previous studies predicted that SARS-CoV SUD forms a tight dimer for G4 binding, with its G4-binding site formed between SUD monomer[15,17].

## Identification of compounds binding to SARS-CoV-2 SUD-core with high affinity

Because SUD plays essential roles in CoV replication through binding to important viral and host proteins and RNAs, disrupting the SUD-mediated biology may provide a feasible antiviral strategy. To explore this, we employed BLI approach to screen two

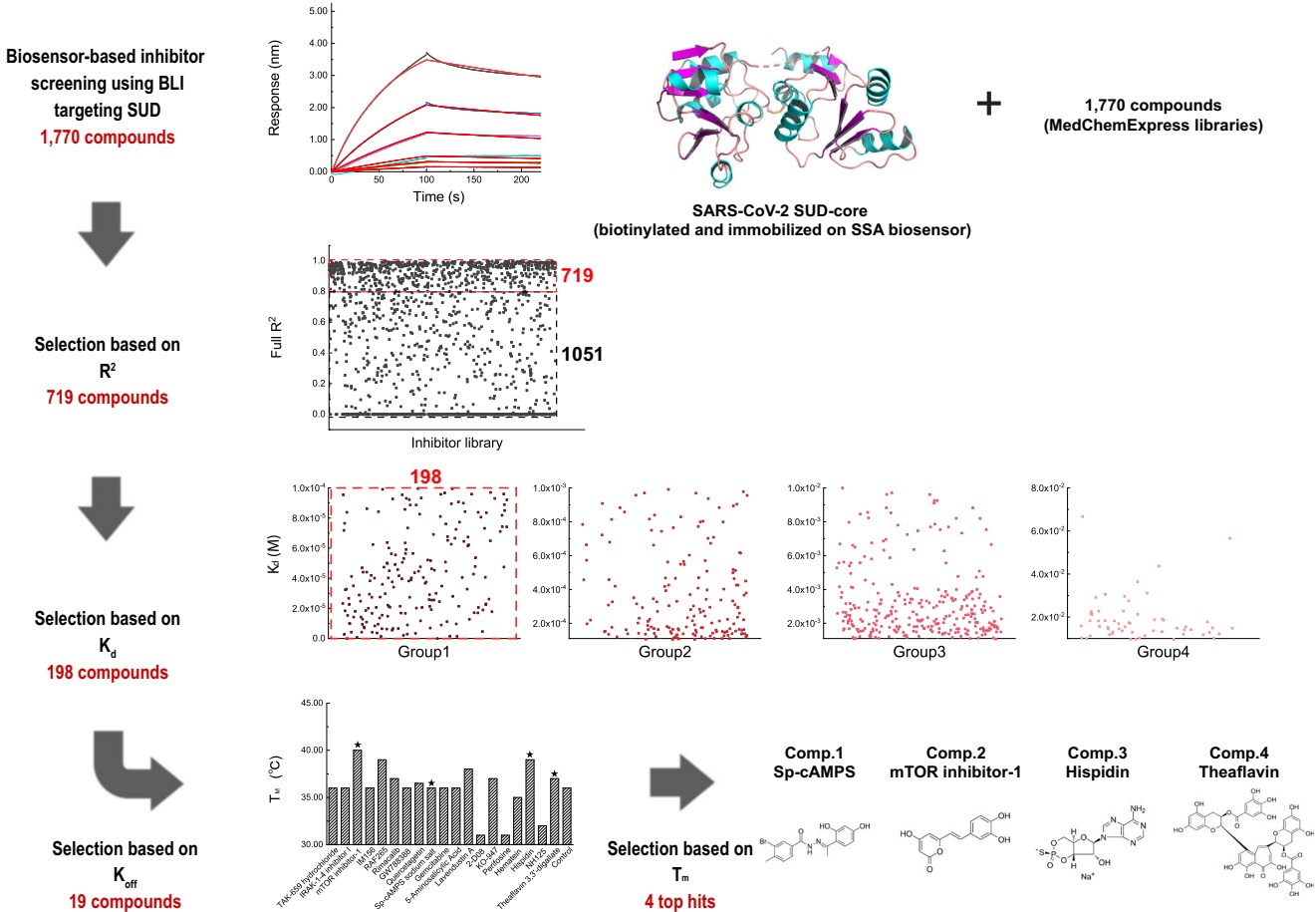

**Fig. 4 | Screening high-affinity compounds binding to SARS-CoV-2 SUD-core using BLI.** MedChemExpress libraries (1770 compounds) were screened for binding with SARS-CoV-2 SUD-core using BLI experiments. Biotinylated SUD-core was immobilization on the SSA sensors. Compounds were diluted in the black 96-well plates to 100 μM in PBS pH = 7.4 for binding with the immobilized SUD-core. Coefficient of determination (full $R^2$) and binding kinetic parameters ($K_d$, $k_{on}$, and $k_{off}$) of each SUD-compound interaction were recorded. 1051 compounds were excluded based on coefficient of determination. The residual 719 compounds were divided in four groups based on $K_d$ values; 198 compounds in group 1 with low $K_d$ was selected for next selection round. Finally, compounds exhibiting low $k_{off}$ and conferred high melting temperature of SUD-inhibitor complexes were selection, which yielded four top hits: Sp-cAMPS sodium salt (Comp. **1**), mTOR inhibitor-1 (Comp. **2**), Hispidin (Comp. **3**) and Theaflavin 3,3'-digallate (Comp. **4**).

commercial libraries totaling 1770 compounds (Nucleotide Compound and Kinase Inhibitor Libraries) for binding to SARS-CoV-2 SUD-core (Fig. 4). As SUD interacts with RNAs and DNAs, our choice were due to richness of the nucleotide-based compounds in these libraries, which may bind SUD and disrupt the SUD-nucleic acids binding.

We prepared biotinylated SARS-CoV-2 SUD-core for immobilization on the Super Streptavidin (SSA) biosensor surface and characterized its binding kinetic with the compound libraries in a high-throughput manner. In the primary screen, we selected 719 compounds based on the coefficient of determination (COD), which is an estimate of the goodness of the curve fitting. The secondary screen was based on binding affinity ($K_d$) values, which included 198 compounds. In the third round, we picked 19 compounds with a slow off-rate ($k_{off}$) in dissociation experiments. Finally, we combined the results from thermal shift experiments and accurate $K_d$ values measured under different concentrations (Supplementary Fig. 8) to prioritize four compounds that conferred high thermostability to SUD-inhibitor complexes and exhibited the strongest binding affinity for SUD-core (Fig. 4). The final hits compounds were Sp-cAMPS sodium salt (Comp.**1**), mTOR inhibitor-1 (Comp.**2**), Hispidin (Comp.**3**) and Theaflavin 3,3'-digallate (Comp.**4**). Binding kinetics parameters of these compounds are summarized in Supplementary Fig. 8 and Supplementary Data 1.

## Hit compounds disrupted the SUD-G4 interaction and the SUD-Paip1M interaction

The SUD-G4 interaction is essential for virus replication[13], therefore we tested the ability of the hit compounds to disrupt this interaction. We synthesized a viral and a human G4-RNAs. The viral G4-RNA was derived from the SARS-CoV-2 genome at position 24268 (5'-GGCUUAUAGGUUUAAUGGUAUUGG-3'), designated G4-24268. This was predicted to be one of the most stable quadruplexes forming sequences in the SARS-CoV-2 genome[19]. The human G4-RNA was derived from a G4 motif located in the 5' untranslated region of telomeric repeat binding factor 2 (TRF2) mRNA[20] (5'-CGGGAGGGCGGG-GAGGGC-3'), designated G4-TRF2.

We first investigated the binding of SARS-CoV-2 SUD-core to viral and host G4-RNAs in the presence of each of the four hit compounds using the electrophoretic mobility shift assay (EMSA, Fig. 5A, B). In the absence of hit compounds, the slow migrating bands corresponding to SUD-G4-24268 and SUD-G4-TRF2 complexes were clearly visible. In the presence of hit compounds (molar ratio of SUD:compound=1:10), whereas Comp.**1** and Comp.**2** had a negligible impact on the SUD-G4 complex, Comp.**3** and Comp.**4** diminished the formation of SUD-G4 complex as evidenced by the increase of the free G4-RNA species in the respective lane was increased accordingly, indicating the disruption of the SUD-G4 complex. In particular, Comp.**4** was the most potent compound in disrupting the SUD-G4 interaction.

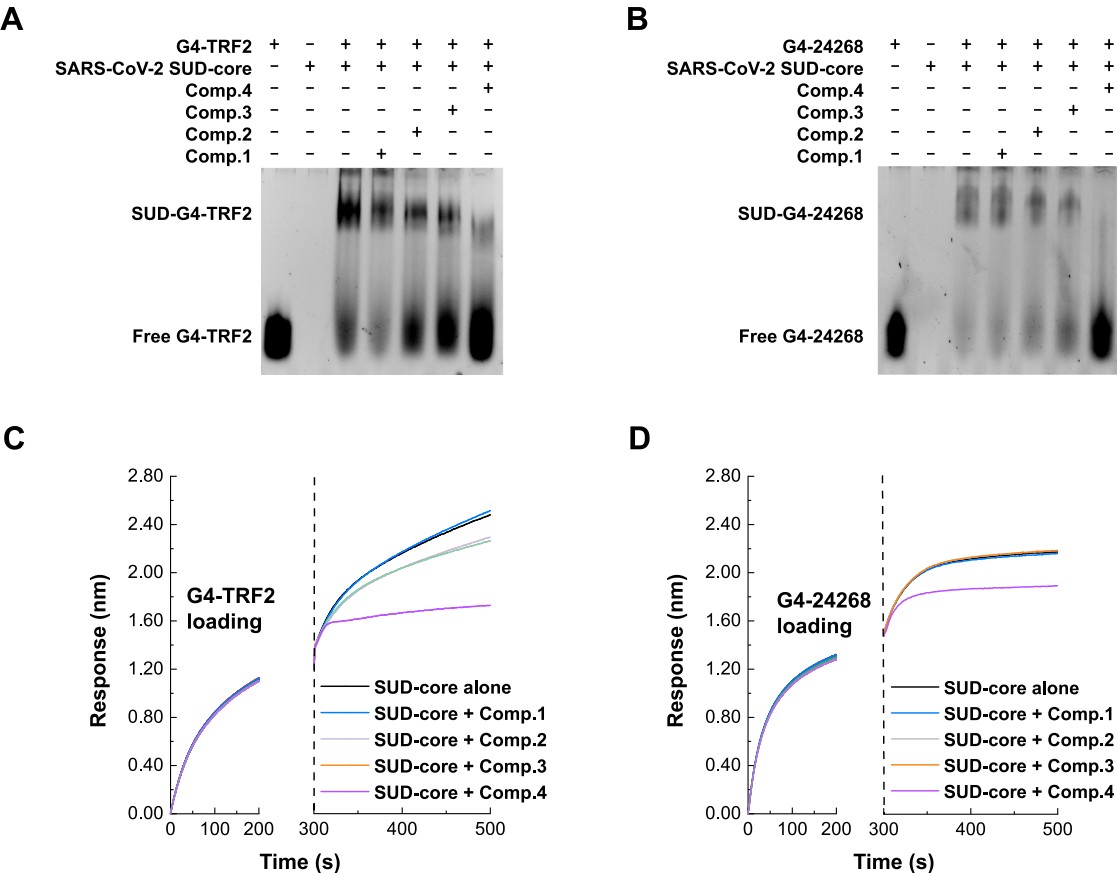

**Fig. 5 | Hit compounds impaired SUD-G4-RNA interaction. A, B** EMSA results showing the binding of viral G4-24286 or host G4-TRF2 with SARS-CoV-2 SUD-core was impaired by hit compounds. SARS-CoV-2 SUD-core was incubated with indicated hit compounds (Comp. **1–4**) before adding G4-TRF2 (panel **A**) or G4-24286 (panel **B**). The mixtures were dissolved using Native-PAGE and stained with SYBR-Gold. **C, D** BLI experiments demonstrating the SUD-G4-RNA interaction was disrupted by hit compounds. Biotinylated G4 RNAs were loaded to SA biosensors for measuring the binding of G4-TRF2 (panel **C**) or G4-24286 (panel **D**) with SARS-CoV-2 SUD-core in the presence of indicated hit compounds. BLI responses during association experiments were aligned.

Next, we employed BLI technique to investigate the binding kinetics of SARS-CoV-2 SUD-core to G4-RNAs in the presence of the hit compounds (Fig. 5C, D). We loaded biotinylated G4-TRF2 or G4-24286 onto SA biosensors, and association experiments were then carried out by dipping the loaded biosensor into wells containing SARS-CoV-2 SUD-core mixed with hit compounds (molar ratio = 1:10). Comp.2 and Comp.3 impaired the association of the SUD-core with host G4-TRF2 and Comp.4 impaired the association to a greater extent. In the case of viral G4, only Comp.**4** impaired the association of SUD-core with G4-24268.

Finally, we also investigated whether the hit compounds could interfere with the SUD-Paip1M interaction using BLI titration. Comp.**1, 3**, and **4** exhibited the abilities to impair binding of SARS-CoV-2 SUD-core with Paip1M protein to various extents, among which Comp.4 was the most potent (Supplementary Fig. 9). Collectively, both our EMSA and BLI results indicate that Comp.**4** was the most potent inhibitor for disrupting the SUD-G4 interaction and the SUD-Paip1M interaction.

To reveal the structural basis for Comp.**4** binding to SARS-CoV-2 SUD, we performed numerous co-crystallization and SUD crystals soaking experiments, but all were unsuccessful. Our highest-resolution dataset (1.35 Å resolution) was obtained by co-crystallizing SARS-CoV-2 SUD-core-CC with Comp.**4**, suggesting that it interacted with SUD during crystallization. However, the electron density of TF3 could not be located. To investigate this further, we analyzed the oligomerization state of SARS-CoV-2 SUD-core in the presence of Comp.**4** using size-exclusion chromatography and analytical ultracentrifugation. Both experiments showed that, instead of forming stable complexes, TF3 undermined the monodispersed conformation of SARS-CoV-3 SUD-core

in solution by promoting nonspecific aggregations (Supplementary Fig. 10), which probably prevented the crystallization of the complex.

## Anti-SARS-CoV-2 activity of the hit compounds

To further validate the antiviral potential of the four hit compounds, they were tested in SARS-CoV-2-infected Vero E6-TMPRSS2 cells. Viral copy in the supernatant were determined by reverse transcription–quantitative polymerase chain reaction (RT-qPCR) method to evaluate the antiviral activity of each compound. Firstly, the hit compounds supplied with a wide range of concentrations were tested in the infected Vero E6-TMPRSS2 cells for preliminary antiviral evaluation (Fig. 6A), where Comp.**4** was found to be the most potent inhibitor against SARS-CoV-2. Next, the cell cytotoxicity of Comp.**4** was further examined (Fig. 6B), resulting a $CC_{50}$ of 98.5 μM. Finally, viral load reduction assays showed that Comp.**4** inhibited SARS-CoV-2 replication with an $EC_{50}$ of 5.9 μM (Fig. 6C), leading to a selectivity index (SI = $CC_{50}/EC_{50}$) of 16.7. Given that the potency of an antiviral compound is largely dependent on cell types, we used a disease-relevant lung epithelial line Calu-3 to reevaluate the efficacy of Comp.**4** (Supplementary Fig. 11). The $CC_{50}$ and the $EC_{50}$ of Comp.**4** on Calu-3 cells were 132.1 μM and 5.8 μM, respectively, which are comparable to the cytotoxicity and potency evaluated on the Vero E6-TMPRSS2 cells.

## Discussion

Covid19 vaccines are less effective in immunocompromised individuals and against new variants. To combat the ever-evolving SARS-

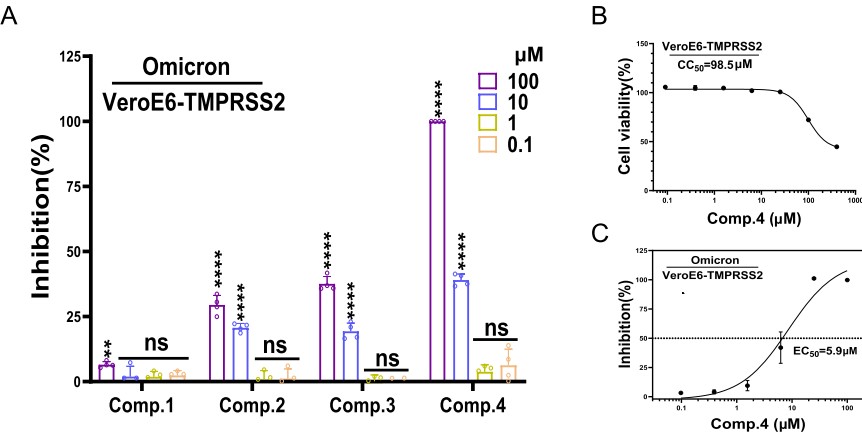

**Fig. 6 | Antiviral activity of the hit SUD inhibitors against SARS-CoV-2. A** Viral load reduction assay of four hit compounds was performed on VeroE6-TMPRSS2 cells using the indicated concentrations. Viral copy in the cell culture supernatant were determined at 48 hpi by RT-qPCR methods and the inhibition of cell proliferation of four compounds were shown. All data were shown as mean ± SEM, $n = 4$ independent experiments. One-way ANOVA for statistical analysis were compared with the lowest compounds concentrations (0.1 μM),

$**P = 0.007539$, $****P < 0.000001$ and ns indicates $P > 0.05$. **B** The cytotoxicity of Comp.4 on VeroE6-TMPRSS2 cells were measured by MTT method using the indicated concentrations. All data were shown as mean ± SEM, $n = 3$ independent experiment. **C** $EC_{50}$ of Comp.4 on VeroE6-TMPRSS2 cells was plotted by a viral load reduction assay. The experiments ($n = 3$) were repeated twice with similar results. All data were shown as mean ± SEM.

CoV-2 virus in the post-vaccine era, antiviral development is important to reduce the impact of infection in people with weak immunity, and to be prepared for the emergence of new highly virulent variants. nsp3 contains multiple domains, many of which are essential for virus replication. In addition to druggability investigation of SARS-CoV-2 nsp3 PLpro and Mac1 domains, we characterized the nsp3 SUD domain structurally and biochemically in this study and provide evidence that SUD contains druggable site for antiviral development. Although SUD lacks enzyme active site, the SUD-G4 interaction is critical for its function; therefore, the G4-binding site on SUD is a probable "hot spot" for screening and drug.

The RNA G4s are present in both virus and host cell, and they play regulatory roles in virus life cycle. Therefore, interfering G4 functions is considered as a promising antiviral strategy. Employing bioinformatic approaches, multiple G4 sequences were identified in SARS-CoV-2 genome as well as cellular mRNAs. An G4 sequence identified in the coding sequence of N protein can be stabilized by G4-specific binders, leading to reduction of N protein level in vivo[21]. The cationic porphyrin compound TMPyP4 binds SARS-CoV-2 G4s and exhibits even better antiviral activity than remdesivir in SARS-CoV-2-infected hamsters[22]. The transmembrane serine protease 2 (TMPRSS2), an essential host factor for SARS-CoV-2 entry, also contains G4s in its mRNAs. G4-specific binders that stabilize the host G4s can attenuate SARS-CoV-2 infection by inhibiting TMPRSS2 translation[23]. In addition to the G4-stablizaion strategies, we provide here evidence that interfering with G4-protein interaction is another valid antiviral strategy.

Although the structure of the SUD-G4 complex has not been experimentally determined, putative G4-binding sites have been predicted for SARS-CoV-2 and SARS-CoV SUD[15,17,20]. The experimentally determined G4 structures are available in the Protein Data Bank, but the protein-G4 complex structure is rare. One of such structures is an RecQ-G4 complex (PDB: 6CRM[24]) in which RecQ protein is bound by an unwound G4 strand rather than a folded G4. While the phosphate backbones of the unwound G4 strand occupies an electropositive groove on RecQ surface comprising several positively charged residues, an aspartate from the guanosine-specific pocket specifically recognizes the 3′-most guanine base of the G4-forming sequence. Therefore, it is likely that both positively and negatively charged residues are important to G4 recognition.

The structural differences between SARS-CoV-2 and SARS-CoV SUD-core imply different modes of G4 recognition. The differences are

at least two folds: (1) the SARS-CoV-2 SUD-core contains loosely linked SUD-N and SUD-M, allowing greater flexibility during G4 binding. By contrast, the disulfide bond linking SARS-CoV SUD-N and SUD-M domains restricts their conformational changes during G4 binding. (2) We did not observe the SARS-CoV SUD-like dimers in the crystal structure of SARS-CoV-2 SUD-core-CC, and PISA software did not identify any other quaternary structures larger than monomers, consistent with the results of size-exclusion chromatography and analytical ultracentrifugation, demonstrating that the SARS-CoV-2 SUD-core formed monodispersed monomers in solution (Supplementary Fig. 10).

The most potent compound identified in this study was Comp.**4** Theaflavin 3,3′-digallate (TF3). We demonstrated that TF3 binds directly to SARS-CoV-2 SUD, disrupting the SUD-G4 interaction, and that it exhibits anti-SARS-CoV-2 (Omicron variant) activity in Vero E6-TMPRSS2 cells. TF3 is an ingredient of black tea, which is well-known for its broad health-benefits, including anti-inflammatory, antioxidant, anticancer and antiviral activities[25]. TF3 targets several viral proteins, such as CoV main proteases[26,27] and the Zika virus NS2B-NS3 protease[28], etc. However, structural evidence for the TF3-target protein interaction remains lacking.

Comparing differences between SARS-CoV-2 and SARS-CoV proteins may help us to identify the molecular determinants underlying virus infectivity, transmissibility and virulence. SUD is unique to Sarbecoviruses, and it was originally considered important for the pathogenesis of these viruses. We identified a key difference between SARS-CoV-2 SUD and SARS-CoV SUD; although the individual domains are similar, their domain linkage, and consequently their domain orientation, differs significantly. The unusual interdomain disulfide bond of SARS-CoV SUD is absent in SARS-CoV-2, which allows greater flexibility between SUD-N and SUD-M. Using reverse genetic system, we found that introducing a disulfide bond into SARS-CoV-2 was lethal to the virus, highlighting the importance of this domain linkage in SUD functions and virus replication.

Disulfide bond formation requires an oxidative environment, but the cytoplasm is typically reductive; therefore, cytosolic proteins rarely require intramolecular disulfide bonds for their functions. However, in case of viral infection, exceptions do occur. Viral infection can cause oxidative stress in the infected cells. Viruses-induced organelle, such as double membrane vesicles (DMV), may contain oxidative microenvironments. An electron cryotomography study revealed that

nsp3 is the major component of a molecular pore complex spanning two membranes of CoV induced DMVs that facilitates transport of RNA into cytoplasm[5]. Although the location of SUD in the molecular pore is unknown, it is possible that an oxidative microenvironment is present there. If the function of the disulfide bond in SARS-CoV SUD is related to oxidative stress or its subcellular location, SARS-CoV-2 SUD must play different roles during viral replication because it has lost the disulfide bond.

In summary, we determined high-resolution structures of SARS-CoV-2 SUD and identified key features that distinguish it from SARS-CoV SUD. The absence of the SARS-CoV SUD intramolecular disulfide bond not only governs SARS-CoV-2 SUD stability, oligomerization, and binding to protein and RNA ligands, but also virus fitness. We identified a hit compound that directly binds with SARS-CoV-2 SUD and disrupts the SUD-G4 interaction, and the potency of the hit compound was confirmed using SARS-CoV-2-infected cells. Our findings identify SUD as a potential target for next-generation anti-SARS-CoV-2 therapeutic development.

## Methods

### Plasmid construction and protein expression
The DNA encoding the middle domain of human Paip1 (Paip1M, residues 78–296) was amplified by polymerase chain reaction (PCR) and was subsequently inserted to pET28a-sumo plasmid between the restriction enzyme sites BamHI and XhoI according to Gao et al.[9]. The genes encoding SARS-CoV SUD-core (nsp3 389-652) and SARS-CoV-2 SUD-core (nsp3 residues 413–676) were synthesized with optimized codon for *E. coli* (Sangon Biotech) and cloned into the pET-Duet-1 vector between the restriction BamHI and HindIII, which expressed the N-terminal 6× His-tag fused SUD-core. The plasmid encoding mutant SARS-CoV-2 SUD-core-CC containing L516C and Y647C and other 17 mutant plasmids as shown in Fig. 3 and Supplementary Fig. 7 were generated by site-directed mutagenesis (QuikChange™). All constructures were verified by DNA sequencing.

The expression of SUD-core proteins and Paip1M follows the similar protocol. Briefly, plasmids were transferred to C3016H competent cells (New England Biolabs). A single colony was picked to inoculate lysogeny broth (LB) medium at 37 °C till a density of OD600 ~ 1.0. Isopropyl-D-1-thiogalactopyranoside (IPTG) was then added to the culture (final concentration ~0.5 mM) to induce the expression and the culturing continued for another 20 h at 18 °C. The bacteria culture was harvested by centrifugation at 2991 × *g* and stored at −80 °C before use.

### Protein purification and crystallization
Bacteria cell pellets of expressing Paip1M were resuspended in lysis buffer A (20 mM Tris-HCl pH 8.5, 500 mM NaCl, and 10 mM imidazole) and disrupted by ultrasonication at 4 °C. Cell debris was removed by centrifugation at 20,000 × *g* for 1 h, and the supernatant was loaded onto Ni-NTA resin. After washing the resin with buffer A, Paip1M protein was eluted by buffer B (20 mM Tris-HCl pH 8.5, 500 mM NaCl and 300 mM imidazole). The 6×His-SUMO tag was cleaved by dialyzing against the dialysis buffer C (20 mM Tris-HCl pH 8.5 and 200 mM NaCl) supplemented with Ulp1 peptidase at 4 °C overnight. The sample was passed through fresh Ni-NTA resin to collect the flow-through containing non-tagged Paip1M protein. Finally, the non-tagged Paip1M was loaded to Superdex 200 10/300 GL column (GE Healthcare) pre-equilibrated with the dialysis buffer C. Paip1M protein was concentrated to ~10 mg/ml and stored at −80 °C before use.

Bacteria cell pellets of SUD-core and all the other SUD-core mutants were resuspended in lysis buffer D containing 1×PBS pH = 7.4 supplemented with 500 mM NaCl, 0.06% 2-Mercaptoethanol and 1 mM phenylmethylsulfonyl fluoride (PMSF); and the cell suspension was disrupted by ultrasonication. Cell debris was removed by centrifugation at 20,000 × *g* for 1 h at 4 °C. The supernatant was loaded

to Ni-NTA resin pre-equilibrated with lysis buffer D. Nonspecifically bound materials were removed by washing the resin with lysis buffer E of 10× bed volume. SUD-core was eluted by elution buffer F containing 1×PBS pH = 7.4, 0.06 % 2-Mercaptoethanol and 300 mM imidazole. The eluate was concentrated using a 10 kDa cut-off centrifugal filter (Millipore). The concentrated sample was loaded to a Superdex 200 10/300 GL column pre-equilibrated with buffer G 1×PBS pH = 7.4. The eluate was analyzed by SDS-PAGE. Fractions containing the target proteins were pooled and concentrate to ~10 mg/ml for crystallization trails.

The crystals of SARS-CoV-2 SUD-core-CC were grown in a buffer H containing 0.2 M lithium sulfate monohydrate, 0.1 M Tris-HCl pH = 8.5 and 25% PEG3350. The crystallization was performed by mixing 1 µl crystallization buffer with 1 µl protein sample in a hanging drop diffusion system at 18 °C. The crystals were soaked in the crystallization buffer H containing 10% glycerol before flash freezing in liquid nitrogen. X ray diffraction data of SARS-CoV-2 SUD-core-CC crystals were collected at the X06DA beamline at the Swiss Light Source, Paul Scherrer Institute, Villigen, Switzerland. Diffraction data was processed using the XDS Package. The crystals diffracted the X ray to 1.35-2.30 Å (Supplementary Figs. 12 and 13). The best diffracting crystals were obtained by adding 10% DMSO and 0.25 mM Comp.**4** in the crystallization drops. SARS-CoV-2 SUD-core-CC structure was determined by molecular replacement using software Phaser[29]. The searching model was SARS-CoV SUD-core structure (PDB 2W2G[15]). Manual model building was conducted using software Coot. The crystal structure was refined using software package Phenix[30] Data collection and refinement parameters are summarized in Table 1.

### Size-exclusion chromatography
Paip1M was mixed with SARS-CoV SUD-core or SARS-CoV-2 SUD-core by molar ratio 1.2:1 and incubated at 4 °C overnight. Subsequently, the mixtures were loaded onto a Superdex 200 column 10/300 GL (GE Healthcare) pre-equilibrated with buffer G. Peaks in chromatograms were analyzed by software Unicron 5.11.

### Electrophoretic mobility shift assay (EMSA)
Purified SARS-CoV-2 SUD-core was mixed with hit compounds by a molar ratio 1:10 at 18 °C in the binding buffer G. G4-RNAs were then added to the mixture by a molar 1:10 molar ratio and the mixture were incubated at 18 °C for 60 min. The resulting mixtures were analyzed by 5% native-PAGE in 1×TBE buffer. The gel was stained by SYBR-Gold (Invitrogen) and photographed using a digital camera equipped with UV light.

### Biotinylation of SARS-CoV-2 SUD-core protein
Target protein was biotinylated by using biotin quick labeling kit (Frdbbio, ARL0020K). Briefly, 1 mg of protein was diluted with adding marking buffer to 1 mg/ml and 10 µl of biotin (10 mM) was added to the sample. The mixture was incubated at 37 °C for 30 min. The unlabeled biotin was removed by 3–4 concentrating-diluting cycles using centrifugal filters.

### Biolayer interferometry (BLI)
The binding of Paip1M with SARS-CoV-2 SUD-core was measured by BLI using ForteBio Octet RED96e Analysis System. Purified Paip1M was immobilized on AR2G biosensors through its amine groups using EDC (1-Ethyl-3-[3-dimethylaminopropyl] carbodiimide hydrochloride)/NHS (N-hydroxysulfosuccinimide) buffer according to the standard amine coupling protocol provided by the manufacture. The loading time of Paip1M (diluted by 10 mM acetate pH 5.0 to 20 µg/ml) to AR2G sensors was 150 s. The biosensors were dipped in purified SUD-core proteins (2-fold serial dilutions from 116.7 µM) in buffer I (1×PBS pH = 7.4, 0.1 % Surfactant p20, 0.1 % bovine serum albumin, 500 mM NaCl) for

association 30 s. The biosensors were then transferred to fresh buffer I for disassociation 80 s.

The binding of G4-RNAs with SARS-CoV-2 SUD-core was measured in buffer I Biotinylated G4-RNAs (200 nM) were loaded on SA biosensor for 200 s. The loaded biosensors were dipped into SUD-core solutions (2-fold serial dilutions from 200 μM) for 50 s; and the biosensors were transferred to fresh buffer I for dissociation 100 s.

MedChemExpress Nucleotide Compound (Cat. No.: HY-L044) and Kinase Inhibitor (Cat. No.: HY-L009) libraries containing total 1,770 compounds were screened for binding with SARS-CoV-2 SUD-core using BLI. All compound powders were dissolved in 100 % dimethyl-sulfoxide (DMSO) to prepare 10 mM stock solutions. SUD-core protein was biotinylated using a biotinylation kit (Frdbbio, ARL0020K) for immobilization on the SSA biosensors specialized for small molecule-protein interactions. 50 μg/ml biotin-SARS-CoV-2 SUD in buffer G was used for loading to SSA biosensor for 1,800 s. All compounds were diluted to 100 μM in buffer G to make sure the final concentration of DMSO is 1.0 % for binding with the immobilized SUD-core protein on SSA biosensor. After association for 150 s, dissociation was carried out in in fresh buffer J (1×PBS pH = 7.4, 1.0 % DMSO) for 180 s.

SUD-core-G4-RNA disruption assay was carried out using BLI. Biotinylated G4-RNAs were synthesized and loaded to SA biosensors for measuring binding with SARS-CoV-2 SUD-core protein in the presence and absence of hit compounds. BLI responses during association were recorded and compared. To avoid the nonspecific binding, extra SA sensors were used for double reference subtraction.

SUD-Paip1M disruption assay was carried out using BLI. Paip1M was biotinylated using a biotinylation kit (Frdbio, ARL0020K); 50 μg/ml biotin-Paip1M were loaded to SA biosensors for measuring binding with SARS-CoV-2 SUD-core protein in the presence and absence of hit compounds. BLI responses during association were recorded and compared. To avoid the nonspecific binding of Compound, extra SA sensors were used for double reference subtraction.

The binding affinity of SUD-core mutants with G4-TRF2 was measured by BLI in buffer I as described before. Briefly, G4-TRF2-loaded biosensors were dipped into 2 μM SUD-core mutants for 50 s and the biosensors were transferred to fresh buffer J for dissociation 100 s.

All BLI experiment were carried out at 25 °C. Data were recorded and analyzed using software ForteBio Data analysis v 11.1. The data was processed with the reference well subtraction and Global fitting with 1:1 model.

### Circular dichroism (CD) spectroscopy

Solution structure and thermostability of SUD proteins were measured by CD spectroscopy as described previously[31]. SUD proteins were diluted to 11 μM in 1×PBS pH = 7.4 and the CD spectra was acquired on Jasco spectropolarimeter (model J-815) by using a 1 nm bandwidth with a 1 nm step resolution from 195 to 270 nm at 20 °C. The final spectra were obtained by subtracting a buffer blank. Thermostability was measured at 222 nm by real-time monitoring the ellipticity change from 20 to 98 °C at a rate of 2 °C/min and the melting temperature (Tm) was analyzed and obtained by software Origin.

### Thermal shift assay

1×SYPRO Orange (Invitrogen) 0.15 μl, 0.6 μl SUD-core protein (0.2 mg/ml) and 29.25 μl 1×PBS (pH = 7.4) were mixed in white bottom Multiwell plates 96 (Roche). The plates were sealed with highly transparent optical-clear quality sealing tape (Roche) and centrifuged at 4 °C at 2000 × g for 1 min before experiments. The plates were heated in an CFX96 Real-Time System (Bio-Rad) from 4 to 95 °C with increment of 1 °C per minute. Fluorescence changes (excitation 470 nm, emission 570 nm) in each well were reordered in real-time. The data was analyzed by Bio-rad CFX Manager 3.0 software to calculate melting temperatures.

### Cell lines and SARS-CoV-2 viruses

Vero E6-TMPRSS2-cells were maintained in DMEM Medium (gibco) containing 10 % FBS and 1 mg/ml G418 Sulfate and 1% Penicillin-Streptomycin. The Calu-3 cell line was purchased from ATCC (HTB-55), and maintained in Dulbecco's Modified Eagle Medium/Nutrient Mixture F-12 (DMEM/F-12) supplemented with 10% of Fetal Bovine Serum (FBS) and 1% of 5000 units/ml Penicillin and 5000 μg/ml. The WT SARS-CoV-2 (HKU-001a strain, GenBank accession number MT230904) and SARS-CoV-2 B.1.1.529/Omicron (GISAID accession number EPI_ISL_7138045) strains were isolated from respiratory tract specimens of laboratory-confirmed COVID-19 patients in Hong Kong and cultured in Vero E6-TMPRSS2 cell[32].

### Antiviral and cytotoxicity evaluation

To evaluate hit compounds cytotoxicity, Vero E6-TMPRSS2 and Calu-3 cells were seeded on a 96-well plate at the concentration of $2 \times 10^4$ per well and incubated with the compounds at different concentrations (2-fold serial dilution from 200 μM) for 48 h. The cytotoxicity of the compounds was determined using an MTT (3-[4,5-dimethylthiazol-2-yl]−2,5 diphenyl tetrazolium bromide) assay as we previously described[33].

To examine the antiviral activity of hit compounds, viral load reduction and/or plaque reduction assays were performed as we described previously[34]. For viral load reduction assay, cells were incubated with DMEM medium containing compounds at different concentrations (2-fold serial dilution from 100 μM) for 1 h, followed by SARS-CoV-2 infection at a multiplicity of infection (MOI) of 0.1. After another 2 h, the infectious inoculate were removed and supplemented with freshly prepared DMEM medium containing hit compounds. The viral copy in the cell culture supernatant was detected at 48 hpi by qRT-PCR methods.

### Generation of recombinant SARS-CoV-2 with nsp3 mutations

Reverse genetic to generate recombinant SARS-CoV-2 was performed as we previously described[35–38]. Both or single Leucine at positions 516 and Tyrosine at positions 647 in nsp3 of SARS-CoV-2 genome were mutated to the cysteine, and recombinant SARS-CoV-2 with nsp3-L516C, nsp3-Y647C and nsp3-L516C-674C were generated in pBAC-SARS-CoV-2 using the homologous recombination, followed by transfection to BHK21-ACE2 cells for 6 h. Then the cells were trypsinized and cell culture supernatant were co-cultured with Vero E6-TMPRSS2 cells for 24 h and 48 h. Vero E6-TMPRSS2 cells culture supernatant were collected every 24 h after co-culture for detection of progeny virus.

### Plaque assay

The confluent Vero E6-TMPRSS2 cells in 24-well culture plates were infected with 10-fold diluted SARS-CoV-2-nsp3-Y647C or SARS-CoV-2, respectively. Cells were incubated with virus supernatant diluted with DMEM for 1 h, and then overlaid with 1% UltraPure™ Low Melting Point Agarose (Thermo Fisher Scientific) in DMEM. At 2 days post-infection (dpi), the cells were fixed with 4% paraformaldehyde and stained with 0.1% crystal violet to visualize plaques.

## Data availability

The complete sequences of SARS-CoV-2 HKU-001a (GenBank: MT230904), SARS-CoV-2 B.1.1.529/Omicron (GISAID: EPI_ISL_7138045), SARS coronavirus Tor2 (GenBank: YP_009944368.1), SARS-CoV-2 WUHAN (GenBank: QIU82068.1), SARS-CoV-2 B.1.1.7 (GenBank: UKQ11044.1), SARS-CoV-2 Beta B.1.351 (GenBank: QWW93434.1), SARS-CoV-2 delta B.1.617.2 (GenBank: QYM 89679.1), Bat coronavirus isolate BANAL-20-52/Laos/2020 (GenBank: MZ937000.1), Bat coronavirus isolate BANAL-20-103/Laos/2020 (GenBank: MZ937001.1), Bat coronavirus isolate BANAL-20-236/Laos/2020 (GenBank: MZ937003.), Bat coronavirus RaTG13 (GenBank: QHR63299.2), Bat coronavirus RacCS203 (GenBank:

QQM18863.1), Pangolin coronavirus (GenBank: QVT76605.1), Bat SARS-like coronavirus WIV1 (GenBank: AGZ48830.1), Bat cov Rc-o319 Japan (GenBank: BCG66625.1), Beta coronavirus sp. RsYN04 (GenBank: QWN56241.1), Bat-SL-CoVZC45 (GenBank: MG772933.1), Pangolin-CoV-GD (GenBank: QIG55944.1), Pangolin-CoV-GX strain (GenBank: QIQ54047.1), Bat SARSr CoV RmYN02 (GISAID: EPI_ISL_412977), Bat SARSr CoV RshSTTT182 (GISAID: EPI_ISL_852604) and Bat SARS coronavirus HKU3-1 (GenBank: AAY88865.2), are available on GenBank and GISAID. All the compounds in this article were purchased from MedChemExpress LLC and the purity of them were all greater than 95% as shown on the MCE official website: https://www.medchemexpress.cn. All accession codes used in this study including 2W2G (Human SARS coronavirus unique domain); 6YXJ (Crystal structure of SARS-CoV macrodomain II in complex with human Paip1); 6CRM (Crystal Structure of RecQ catalytic core from C. sakazakii bound to an unfolded G-quadruplex) are available in the PDB protein databank. The atomic coordinates and structure factors have been deposited in the Protein Data Bank under the accession codes: 8GQC (SARS-CoV-2 SUD-core-CC Soaked with Comp.4) and 8HBL (SARS-CoV-2 SUD-core-CC Protein alone). Source data are provided as a source data file. Source data are provided with this paper.

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

## Acknowledgements

We thank the staff of PX III beamline at the Swiss Light Source, Paul Scherrer Institute (Villigen Switzerland) for assistance in data collection. We thank the staffs of BL19U1 beamline of National Facility for Protein Science in Shanghai (NFPS) for assistance in data collection. We thank the staffs from the Core Facility of Institute of Pathogen Biology, Chinese Academy of Medical Sciences. This work was supported by the Chinese Academy of Medical Sciences (CAMS) Innovation Fund for Medical Sciences (2022-I2M-1-021); National Natural Science Foundation of China/RGC Joint Research Scheme (No. 82261160398, N_HKU767/22); the National Research Program Covid-19 (NRP78) from the Swiss National Science Foundation (grant number 4078P0_198290); the European Union's Horizon 2020 research and innovation program under the Marie Skłodowska-Curie grant agreement (No. 884104 PSI-FELLOW-III-3i); Fundamental Research Funds for the Central Universities (3332021092); Theme-Based Research Scheme of the Research Grants Council (T11-709/21-N); The Government of the Hong Kong Special Administrative Region; Guangdong Natural Science Foundation (2023A1515012907).

## Author contributions

B.Q., S.Y., and S.C. designed the study and wrote the paper. B.Q., Z.L., K.T. performed experiments. S.A. and M.W. collected the crystal data. S.C. and B.Q. determined the structures. B.Q. and Z.L. performed protein purification and biochemical experiments. K.T. and T.W. performed virus rescue and intracellular inhibitor evaluation. B.Q., Z.L., K.T., T.W., Y.X., S.Y., and S.C. analyzed the data and revised the paper. All authors reviewed the results and approved the final version of the manuscript.

## Competing interests

The authors declare no competing interests.
