## [Peer Review File · Nature Communications]

REVIEWER COMMENTS

Reviewer #1 (Remarks to the Author):

- (1) The authors reported the 3D structure of SUD of SARS-CoV-2 and provided evidence that SUD of SARS-CoV-2 could be drug target with biolayer interferometry and cell-based assays. This manuscript could be accepted with minor revision.
- (2) Even though the subcellular location of nsp3 is in the reductive cytoplasm, after the lysis and release and the purification, the protein would be exposed to the oxygen and then form the disulfide bond automatically. And the disulfide bond might restrain the conformational flexibility of SUD domain of nsp3, which might be essential for its biological activity. So it is better to have a solution-based structural assay like CD or SAXS to assist the data from the crystal structures.
- (3) From the international academic community, NSP3 should be nsp3.
- (4) The engineered disulfide bond connects M and N region and restrain the conformation. Did the authors try to use DTT or TCEP to reduce the engineered disulfide bond and then do some assays, such as CD or cell-based assays?
- (5) Did the authors try to get the complex structure of SUD bound to the hit compounds?
- (6) For the data collection and refinement table, 120.00 should be 120
- (7) For the resolution, the parantheses should include the highest shell.
- (8) The B-factor should have a unit, not just a number.
- (9) In the figure S5, 3 and 6 and 10 shows the binding data, it is not fitted well and thus the Kd is not quite reliable.
- (10) What is the possible explanation for the disruption of the monodispersity of SARS-CoV-2 SUD-core by cmp4?

Reviewer #2 (Remarks to the Author):

Qin et al. present an interesting study on the SARS-unique domain (SUD) of SARS-coronavirus-2 (SARS-2). This domain of non-structural protein 3 (nsp3) shares 75% sequence identity with the corresponding SUD of SARS-CoV (SARS-1), but it lacks the disulfide bond connecting subdomains N and M in the latter. Interestingly, when the authors engineered this disulfide bond into the appropriate position in the SARS-2 SUD, the virus was no longer viable.

SARS-1 SUD has previously been shown by the Hilgenfeld group to bind G-quadruplexes (G4) and the present authors confirm this for SARS-2 SUD. They also show that the interaction between SUD and the G4 is interrupted by theaflavin-3,3'-digallate (TF3), which binds to SARS-2 SUD with an Kd of 2.8 μ M. TF3 exhibited antiviral activity in SARS-CoV-2-infected Vero E6 cells (EC50 = 5.9 mM) and was not toxic. These results suggest that SUD may be a druggable protein.

This excellent manuscript describes important work and I have only a few comments:

Line 94: "Antiviral agent development has largely focused on NSP3 since the beginning of COVID-19 pandemics". I do not think that this statement is true. Targets most in focus were (and are) the Mpro (Nsp5) and the RdRp (Nsp12), whereas most efforts on Nsp3 have targeted the PLpro.

The following reference to early work on SUD is missing: J. Tan, Y. Kusov, D. Mutschall, S. Tech, K. Nagarajan, R. Hilgenfeld & C.L. Schmidt: The "SARS-unique domain" (SUD) of SARS coronavirus is an oligo(G)-binding protein. *Biochem. Biophys. Res. Commun.* 364, 877-882.

Line 197: "... and diffracted the X rays to 1.35-2.30 Å resolution": This is potentially slightly confusing, because the resolution range in diffraction data collection is commonly given in the format "from ... to". What is meant here though is that some crystals diffracted to 1.35 Å and others only to 2.3 Å.

Line 200: Please provide the full reference, not just the PDB code.

Line 218: R.M.S. difference values should be provided for the individual domains, SUD-N and SUD-M.

Lines 220 – 230: The usage of AlphaFold2 does not add much to the quest for the “true” structure here.

Reviewer #3 (Remarks to the Author):

Qin and colleagues investigate structural and functional differences of the Nsp3 SUD domain from SARS-CoV and SARS-CoV-2. While a structure of SARS-CoV Nsp3 SUD has been previously solved, none exist for SARS-CoV-2 Nsp3 SUD. Interestingly, SARS-CoV SUD contains a disulfide bond linking the SUD-N and SUD-M domains. This disulfide bond is absent from SARS-CoV-2 Nsp3 SUD as well as many closely related sarbecoviruses and the authors speculate that this may result in differences in both structure and function. Being unable to obtain a crystal structure of SARS-CoV-2 Nsp3 SUD, the authors introduce a disulfide bond mimicking that of SARS-CoV Nsp3 SUD, reasoning it would limit domain flexibility to facilitate crystal formation, which resulted in the authors solving a high-resolution structure of SARS-CoV-2 Nsp3 SUD. The newly solved structure of SARS-CoV-2 Nsp3 SUD provides novel structural insights into differential function between SARS-CoV and SARS-CoV-2. Namely, the absence of a disulfide bond of SARS-CoV-2 suggests higher flexibility between the distinct domains of SUD. Indeed, the authors demonstrate differential thermal stability and interaction with the host factor Paip1M as well as potential differential modes of binding to Paip1M. Further, the authors use their structure to predict and validate charged amino acids that are important for interacting with G4 residues. The authors then conduct a BLI-based high-throughput screen to identify compounds that bind to SARS-CoV-2 Nsp3 SUD and identify 4 high binders. Two compounds in particular demonstrate the capacity to interfere with an interaction between Nsp3 SUD and G4 complexes and possess modest antiviral activity *in vitro*. In sum, the authors use a creative way to solve a structure of SARS-CoV-2 Nsp3 SUD, although the introduction of a disulfide bond does cast doubt on how similar this structure is to the native SARS-CoV-2 structure. That said, this was an enjoyable manuscript to read and the authors do provide novel insight into differences between SARS-CoV-2 vs. SARS-CoV Nsp3 SUD. A few additional controls are suggested below that would further confirm the authors findings and strengthen the manuscript.

Major:

1. For all functional assays (melting curve, G4 interaction, and especially Paip1M interaction) it is important for the authors to compare four proteins, SARS-CoV-2 SUD with and without their CC and also SARS-CoV SUD with and without CC (assuming the SARS-CoV SUD is stable without its native disulfide bond). What's unclear to me is if this disulfide bond is the major player that differentiates CoV vs. CoV-2 SUD or if other differential amino acids are also at play?
2. It was unclear to this reviewer how the authors conclude that the mode of binding of Paip1M to SARS-CoV vs. SARS-CoV-2 Nsp3 SUD was distinct? Could the authors clarify this point?
3. It may improve the flow of the manuscript if the authors move the description of figure 6 (and S6) from the discussion and into the results section. It would also be great to provide a description and citation for why charged amino acids contribute to G4 binding.
4. In addition to testing the capacity of candidate compounds to interfere with G4 binding, could the authors also determine if they interfere with the interaction with Paip1M?
5. It is important to assess the relative stability of the mutants produced in Figure 6. Are they all equally stable to the WT protein?
6. In Figure S6A SEC, how large is compound 4? Is the retention time of this complex consistent with the additive size of both molecules? Are the proteins are coming off the column in the void volume?

7. Critically, large differences in antiviral compound potency have been observed in distinct cell types. Vero-E6/TMPRSS2 are a highly artificial and non-relevant cell type for testing antiviral compounds against SARS-CoV-2. Please use a lung epithelial line such as Calu-3 to test for efficacy of candidate compounds against SARS-CoV-2. Primary human lung cells would be ideal but much more difficult to work with.

Minor

1. Please show plaque assays for the SARS-CoV-2 dead mutants (Figure 1).

2. Could the authors please expand their discussion to include more information of the role G4 structures are known to play in the SARS-CoV-2 lifecycle? What is known about the role of charged amino acids in interaction with G4 structures?

3. Figure S5 please indicate what the concentrations of dilutions of compounds.

4. Fig. 4C/D (and throughout the manuscript) the color scheme used in figures are not color blind friendly and thus make it difficult to differentiate conditions. Could the authors please use an alternative color scheme?

Reviewer #1 (Remarks to the Author):

(1) The authors reported the 3D structure of SUD of SARS-CoV-2 and provided evidence that SUD of SARS-CoV-2 could be drug target with biolayer interferometry and cell-based assays. This manuscript could be accepted with minor revision.

Response: Thank you very much for the positive comments.

(2) Even though the subcellular location of nsp3 is in the reductive cytoplasm, after the lysis and release and the purification, the protein would be exposed to the oxygen and then form the disulfide bond automatically. And the disulfide bond might restrain the conformational flexibility of SUD domain of nsp3, which might be essential for its biological activity. So it is better to have a solution-based structural assay like CD or SAXS to assist the data from the crystal structures.

Response: We agree that a solution structure-based method is necessary to validate our crystallographic findings. We therefore carried out circular dichroism (CD) assays for SARS-CoV-2 SUD-core, SARS-CoV-2 SUD-core-CC and SARS-CoV SUD-core under the oxidative conditions. The results show that the spectrum of SARS-CoV-2 SUD-core-CC is highly different from that of SARS-CoV-2 SUD-core, although they are only different in two disulfate bridge forming residues; nevertheless, the spectrum of SARS-CoV-2 SUD-core-CC is more similar to that of SARS-CoV SUD-core that contains a natural disulfate bridge. We also measured melting temperatures (T_m) for these proteins. SARS-CoV-2 SUD-core exhibits lower T_m than the other two proteins, suggesting the disulfate bridge within SUD contributes to overall SUD stability in solution (see Figure below).

We added a new paragraph in the main text (line 233) to describe these results. A new Supplementary Fig. 4 illustrating the CD spectroscopy results is included in the revised supplementary materials.

(3) From the international academic community, NSP3 should be nsp3.

Response: We have changed 'NSP3' to 'nsp3' throughout the text.

(4) The engineered disulfide bond connects M and N region and restrains the conformation. Did the authors try to use DTT or TCEP to reduce the engineered disulfide bond and then do some assays, such as CD or cell-based assays?

Response: According to this reviewer's suggestion, we measured CD spectra for SARS-CoV-2 SUD-core-CC under reductive (by adding DTT 2mM) and oxidative conditions respectively (Figure below).

The spectrum of the DTT treated sample showed slight difference from that of the untreated sample (oxidative conditions). Our guess is that DTT is probably insufficient to revert the already formed S-S bridge to the reductive SH state under our experimental conditions. We also tried TCEP as the reducing reagent, but TCEP caused heavy precipitation of our SUD samples.

(5) Did the authors try to get the complex structure of SUD bound to the hit compounds?

Response: Yes, the authors of this paper tried to co-crystallize SUD-compound complexes for over a year. Four hit compounds from our BLI screening were all tested in co-crystallization. Soaking compounds with apo SUD crystals were

also performed. We collected over a hundred datasets but were unable to observe the compound densities. As shown in Table 1, we report two SUD crystal structures here. The crystal structure with the highest resolution was obtained from Comp.4 (TF4) soaked SUD crystals, suggesting Comp.4 interacted with SUD and led to optimization of the crystal packing; unfortunately, we did not find electron densities for Comp.4.

These observations are explained in the main text (line 388) : “ Our highest resolution dataset (1.35 Å resolution) was obtained by co-crystalizing SARS-CoV-2 SUD-core-CC with TF3, suggesting that the binding of TF3 with SUD during crystallization. However, the electron density of TF3 could not be located.”

(6) For the data collection and refinement table, 120.00 should be 120

Response: Thanks for pointing this out, we have revised the digit accordingly..

(7) For the resolution, the parantheses should include the highest shell.

Response: We have added the highest resolution range in Table 1, please check the revised Table 1.

(8) The B-factor should have a unit, not just a number.

Response: We have added the unit for B-factor in Table 1, please check the revised Table 1.

(9) In the figure S5, 3 and 6 and 10 shows the binding data, it is not fitted well and thus the Kd is not quite reliable.

Response: Thanks for the comments. We have optimized the experimental conditions and repeated the BLI measurements for Comp.3 and Comp.6. Please refer to the new Supplementary Fig.8 for BLI results, and the data fitting is improved. We reprocessed the BLI results for Comp.10 by omitting the highest concentration data. In the revised Fig.S8, the full R² values (ranges 0-100% rating curve fitting quality) of the three BLI measurements are all above 80%: Comp.3 96.54%, Comp.6 82.57% and Comp.10 99.09%, indicating reasonable fitting qualities. Revised Kd values and other binding kinetic parameters are updated in revised Supplementary Table 2.

It is worth noting that the Kd of Comp.10 reduced after data reprocessing, suggesting higher binding affinity. The reason why Comp.10 was not selected is because Comp.10 exhibited the lowest Tm in thermal shift experiments, we do not consider it as a good hit compound (Supplementary Table 2).

(10) What is the possible explanation for the disruption of the monodispersity of SARS-CoV-2 SUD-core by comp4?

Response: Supplementary Fig. 6 suggests that addition of Comp.4 induced SUD aggregation. The size-exclusion chromatography profile shows the SUD-Comp.4 complex eluted from the void volume, suggesting the formation of SUD aggregations. The AUC experiment reveals several huge species after addition of Comp.4, which is consistent with size-exclusion results. According to five general protein aggregation mechanisms (PMID: 19519409), comp.4 might have induced SUD aggregation through the mechanism 2: aggregation of conformational altered or partially disordered monomers. The binding of Comp.4 with SUD probably drives SUD monomer into a non-native state and partially disorders the protein; the altered SUD monomer interacted with each other strongly and very likely formed irreversible aggregates. Aggregated SUD likely loses its function during virus replication.

Reviewer #2 (Remarks to the Author):

(1) Line 94: "Antiviral agent development has largely focused on NSP3 since the beginning of COVID-19 pandemics". I do not think that this statement is true. Targets most in focus were (and are) the Mpro (Nsp5) and the RdRp (Nsp12), whereas most efforts on Nsp3 have targeted the PLpro.

Response: We agree and have rewrote the sentence in the main text (Line 94).

(2) The following reference to early work on SUD is missing: J. Tan, Y. Kusov, D. Mutschall, S. Tech, K. Nagarajan, R. Hilgenfeld & C.L. Schmidt: The "SARS-unique domain" (SUD) of SARS coronavirus is an oligo(G)-binding protein. Biochem. Biophys. Res. Commun. 364, 877-882.

Response: Yes, we have added this reference as number 18th reference and inserted the citation in the main text (line 122).

(3) Line 197: "... and diffracted the X rays to 1.35-2.30 Å resolution": This is potentially slightly confusing, because the resolution range in diffraction data collection is commonly given in the format "from ... to". What is meant here though is that some crystals diffracted to 1.35 Å and others only to 2.3 Å.

Response: We apologize for the ambiguous expressions here. We measured more than a hundred of SUD crystals, hoping to improve the diffraction and

obtain the SUD-compound complex structures. The initial crystals of SARS-CoV-2 SUD-CC diffracted to lower resolutions (~ 2.3 Å); later crystals were improved and diffracted to higher resolution (~ 1.6 Å). Finally, the SUD crystals soaked with Comp.4 diffracted to even better 1.35 Å. We added descriptions of our SUD crystallization process in line 198.

(4) Line 200: Please provide the full reference, not just the PDB code.

Response: We have inserted reference 15 after PDB 2W2G in line 200 and inserted reference 16 after PDB 6YXJ in line 820.

(5) Line 218: R.M.S. difference values should be provided for the individual domains, SUD-N and SUD-M.

Response: We have provided the RMSD values for individual SUD-N and SUD-M in Line218. These data are also provided in Supplementary Table 1.

(6) Lines 220 – 230: The usage of AlphaFold2 does not add much to the quest for the “true” structure here.

Response: Our reasoning of using AlphaFold2 was to predict a natural SARS-CoV-2 SUD core structure without the engineered disulfide bond between SUD-N and SUD-M. The crystallized SUD protein present here is a mutant SARS-CoV-2 SUD core-CC containing an engineering a disulfide bond between SUD-N and SUD-M, which we found essential for its crystallization. By contrast, natural SARS-CoV-2 SUD core did not crystallize, probably due to high flexibility between SUD-N and SUD-M. Therefore, the AlphaFold2 model of natural SARS-CoV-2 SUD may provide clues for its genuine conformation.

Reviewer #3 (Remarks to the Author):

Major:

1. For all functional assays (melting curve, G4 interaction, and especially Paip1M interaction) it is important for the authors to compare four proteins, SARS-CoV-2 SUD with and without their CC and also SARS-CoV SUD with and without CC (assuming the SARS-CoV SUD is stable without its native disulfide bond). What's unclear to me is if this disulfide bond is the major player that differentiates CoV vs. CoV-2 SUD or if other differential amino acids are also at play?

Response: Thanks for the insightful comments. We have therefore preformed

additional experiments to address this reviewer's questions. We carried out BLI titrations for binding of SARS-CoV SUD-core, SARS-CoV-2 SUD-core and SARS-CoV-2 SUD-core-CC with Paip1M protein. The K_d values are compared and illustrated in the revised Fig. 2C and Supplementary Fig. 6. We found that SARS-CoV SUD-core has higher binding affinity than that of SARS-CoV-2 SUD-core. Introducing an S-S bridge to SARS-CoV-2 SUD-core improved the binding affinity, suggesting the S-S bridge is important to SUD-Paip1M interaction. We did not perform experiments for SARS-CoV SUD mutant lacking the natural S-S bridge because the protein was unstable. The main scope of this paper is SARS-CoV-2 SUD.

Next, we tested G4 binding for SARS-CoV SUD-core, SARS-CoV-2 SUD-core and SARS-CoV-2 SUD-core-CC and obtained similar results. SARS-CoV SUD-core bound G4-TRF2 with ~ 10-folds higher affinity than that for SARS-CoV-2 SUD-core, and introducing an S-S bridge to SARS-CoV-2 SUD-core improved G4 binding by ~34-folds. Collectively, although we could not rule out other residues' function, the disulfide bridge form residues play a key role in the SUD functions, including protein and nucleic acid binding.

We added more description of these results in the main text (Line 255). Please note, all BLI sensorgrams are transferred to Supplementary Fig. 6, and column graphs of K_d comparison are added in the revised Fig. 2C.

To further strengthen the findings, we have added CD spectroscopy result for SARS-CoV SUD-core, SARS-CoV-2 SUD-core and SARS-CoV-2 SUD-core-CC in the revised manuscript, which also shown the S-S bridge is important to SUD solution structure and thermal stability (Supplementary Fig. 4).

2. It was unclear to this reviewer how the authors conclude that the mode of binding of Paip1M to SARS-CoV vs. SARS-CoV-2 Nsp3 SUD was distinct? Could the authors clarify this point?

Response: Thank you for the comment. Our structural analyses (described in Line 256-267) suggest that while the SARS-CoV SUD - Paip1M interaction adopts a "induced-fit" mode, SARS-CoV-2 SUD -Paip1M interaction adopts an "lock-and-key" mode. This conclusion is based on: (1) The crystal structure of the SARS-CoV SUD-Paip 1M complex demonstrates that the N-terminal loop (~ 20 residues) of SARS-CoV SUD-N directly interacts with Paip 1M (PubMed: 33876849, PDB: 6YXJ). This loop adopts a different conformation in the absence of Paip 1M (SARS-CoV SUD apo structure, PDB: 2W2G), which is incompatible with binding. Thus, the binding of Paip 1M induces conformational changes of the N-terminal loop, and the binding resembles the "induced-fit" mode. (2) We determined the structure of SARS-CoV-2 SUD-CC in this paper

and found that the N-terminal loop of SARS-CoV-2 SUD-N adopts a similar conformation as that in the Paip 1M bound SARS-CoV SUD, indicating the unbound the N-terminal loop of SARS-CoV-2 SUD-N already forms a binding compatible conformation. These analyses are illustrated in revised Fig. 2D (see below) and described in main text.

D

Taken together, our structural analyses suggest that the SARS-CoV-2 SUD-Paip 1M interaction does not require conformational changes of the SUD N-terminal loop, thus their binding mode resembles the “lock-and-key” mode.

3. It may improve the flow of the manuscript if the authors move the description of figure 6 (and S6) from the discussion and into the results section. It would also be great to provide a description and citation for why charged amino acids contribute to G4 binding.

Response: We agree with this reviewer and have moved the description of the old Fig. 6 to Results section (Line268). To improve the flow of the paper, the original Fig. 6 is reordered to Fig. 3. New section in Result has a subtitle: “Analysis of key residues involved in SUD-G4 interaction”. We added more description of why charged residues contribute to G4 binding, thus were used for mutagenesis. One more citation is also added here.

Description of the old Supplementary Fig. 6 has moved to Line318. The old Supplementary Fig. 6 is reordered to Supplementary Fig. 10.

4. In addition to testing the capacity of candidate compounds to interfere with G4 binding, could the authors also determine if they interfere with the interaction with Paimp1M?

Response: We carried out additional experiment to investigate whether the hit compounds could interfere with the SUD-Paip1M interaction using BLI. The new results show that Comp.1, 3 and 4 exhibited the abilities to impair binding of SARS-CoV-2 SUD-core with Paip1M protein to various extents, among which Comp.4 was the most potent. We also added description of these results in line 318. New BLI titrations were added to Supplementary Fig. 9.

5. It is important to assess the relative stability of the mutants produced in Figure 6. Are they all equally stable to the WT protein?

Response: We added SDS-PAGE results for all SUD mutants used in the new Supplementary Fig. 7. All proteins were expressed and purified to high purity. We also measured T_m values for all mutants using thermal shift assays. The majority of SUD mutant showed similar-to-WT thermal stability. Although there are less stable mutants, but their T_m are well above 25 °C, the temperature we used for BLI experiments. These results demonstrate all SUD mutant were relatively stable under our experimental conditions. Please check more description of these result in Line 348 and a new Supplementary Fig. 7 reporting SDS-PAGE and T_m measurements.

6. In Figure S6A SEC, how large is compound 4? Is the retention time of this complex consistent with the additive size of both molecules? Are the proteins are coming off the column in the void volume?

Response: The molecular weight of Comp.4 (TF3) is 869 Da, the MW of the SUD protein is ~30 KDa; therefore, the theoretical molecular weight of a 1:1 molar ratio complex should be slightly larger than 30 KDa. The gel filtration column employed in this study was Superdex 200 10/300, which has a separation range of 10 KD to 600 KD and the result shown that the complex eluted the column in the void volume. This is probably because Comp.4 induced aggregation of SUD thus undermined its function. Similar question was raised by referee #1, please check our responses to his last question.

7. Critically, large differences in antiviral compound potency have been observed in distinct cell types. Vero-E6/TMPRSS2 are a highly artificial and non-relevant cell type for testing antiviral compounds against SARS-CoV-2. Please use a lung epithelial line such as Calu-3 to test for efficacy of candidate compounds against SARS-CoV-2. Primary human lung cells would be ideal but much more difficult to work with.

Response: As suggested by this reviewer, we re-evaluated anti-SARS-CoV-2

potency of Comp.4 using lung epithelial Calu-3 cells and the result is shown in Supplementary Fig. 11. The new results confirm that Comp.4 has similar cytotoxicity and EC_{50} in the two cell lines tested. We have added more description in Line 330.

Minor

1. Please show plaque assays for the SARS-CoV-2 dead mutants (Figure 1).

Response: All plaque results are shown in the revised Fig. 1C. Please check the new Fig. 1.

2. Could the authors please expand their discussion to include more information of the role G4 structures are known to play in the SARS-CoV-2 lifecycle? What is known about the role of charged amino acids in interaction with G4 structures?

Response: We added more information discussing the role of G4 in SARS-CoV2-2 life-cycle, and provide further information of the role of charged residues in G4-protein interaction in Discussion section in line 342. More references were also added. Please check the revised main text.

3. Figure S5 please indicate what the concentrations of dilutions of compounds.

Response: The concentrations of dilutions of compounds have been added in Supplementary Fig. 8 (original Supplementary Fig. 5)

4. Fig. 4C/D (and throughout the manuscript) the color scheme used in figures are not color blind friendly and thus make it difficult to differentiate conditions. Could the authors please use an alternative color scheme?

Response: Yes, we have changed the color scheme to generate new Fig. 5 (original Fig. 4) in the manuscript.

REVIEWERS' COMMENTS

Reviewer #1 (Remarks to the Author):

The revised manuscript is well written and explained the molecular mechanism of SUD of SARS-CoV2 in details. It could be accepted.

Reviewer #3 (Remarks to the Author):

The authors have addressed all of my comments and provided additional data strengthening their study. I would like to congratulate the authors on a very interesting study.